# Natural diversity screening, assay development, and characterization of nylon-6 enzymatic depolymerization

Elizabeth L. Bell ®[1,2], Gloria Rosetto[1], Morgan A. Ingraham ®[1,2], Kelsey J. Ramirez ®[1,2], Clarissa Lincoln[1,2], Ryan W. Clarke ®[1,2], Japheth E. Gado[1,2], Jacob L. Lilly[3], Katarzyna H. Kucharzyk[3], Erika Erickson ®[1,2] & Gregg T. Beckham ®[1,2] ✉

Successes in biocatalytic polyester recycling have raised the possibility of deconstructing alternative polymers enzymatically, with polyamide (PA) being a logical target due to the array of amide-cleaving enzymes present in nature. Here, we screen 40 potential natural and engineered nylon-hydrolyzing enzymes (nylonases), using mass spectrometry to quantify eight compounds resulting from enzymatic nylon-6 (PA6) hydrolysis. Comparative time-course reactions incubated at 40-70 °C showcase enzyme-dependent variations in product distributions and extent of PA6 film depolymerization, with significant nylon deconstruction activity appearing rare. The most active nylonase, a NylC$_K$ variant we rationally thermostabilized (an N-terminal nucleophile (Ntn) hydrolase, NylC$_K$-TS, $T_m$ = 87.4 °C, 16.4 °C higher than the wild-type), hydrolyzes 0.67 wt% of a PA6 film. Reactions fail to restart after fresh enzyme addition, indicating that substrate-based limitations, such as restricted enzyme access to hydrolysable bonds, prohibit more extensive deconstruction. Overall, this study expands our understanding of nylonase activity distribution, indicates that Ntn hydrolases may have the greatest potential for further development, and identifies key targets for progressing PA6 enzymatic depolymerization, including improving enzyme activity, product selectivity, and enhancing polymer accessibility.

Polyamides (PAs), often referred to as nylons, are versatile, amide bond-linked, petroleum-derived thermoplastics. Considered high-performance materials, PAs are used in a range of applications, including in textiles, fishing nets, packaging, and medical devices[1]. The most prevalent nylons are PA6 and PA6,6, derived from caprolactam or adipic acid and hexamethylene diamine monomers, respectively, with a combined annual global consumption of 7.67 million tons in 2018[2]. The production of PA6 and PA6,6 is both energy- and greenhouse gas (GHG) emissions-intensive, contributing a combined 197 MJ/kg of energy use and 10.4 kg-CO$_2$e/kg GHG emissions for US nylon consumption alone[2]. Furthermore, as with many anthropogenic polymers, PA6 and PA6,6 are sparingly biodegradable, hence they can accumulate in the environment[3]. Circularization of the nylon economy via recycling offers a potential avenue for reducing energy consumption and GHG emissions, decreasing reliance upon fossil fuels, and minimizing environmental impacts of PA waste[4,5]. However, little post-consumer nylon waste is currently collected, and recycling proves challenging as PAs are often combined with other polymers[3,6]. Hence, to complement a number of currently successful, specialized commercial PA recycling operations[7–10], and several promising

[1]Renewable Resources and Enabling Sciences Center, National Renewable Energy Laboratory, Golden, CO 80401, USA. [2]BOTTLE Consortium, Golden, CO 80401, USA. [3]Battelle Memorial Institute, Columbus, OH 43201, USA. ✉e-mail: gregg.beckham@nrel.gov

deconstruction methods described in the literature[11–16], there is need for a wider catalog of nylon recycling approaches to manage this polymer at end-of-life[17], with selective techniques that release PA monomers being of particular interest for closed-loop recycling[18,19].

Chemical recycling using enzymes is an emerging approach with the potential to become a selective method for plastic monomer recovery[5,20]. Excitingly, for poly(ethylene terephthalate) (PET), enzymatic recycling is advancing towards commercial deployment[21,22]. Such successes in biocatalytic PET depolymerization have motivated interest in emulating similar approaches with other polymers, with interests mainly focused on other C–O and C–N linked plastics[23,24], including nylons. Indeed, PAs are a logical choice for enzymatic depolymerization as both amide bonds and amide bond-hydrolyzing enzymes are prevalent in nature. A key first step is the identification of a biocatalyst capable of PA deconstruction, a potentially challenging endeavor as the amide bonds in nylons exhibit limited accessibility due to strong intermolecular hydrogen-bonding networks[25]. Nevertheless, several microorganisms with measurable PA biodegradation activity have been reported[26–36], indicating that enzymatic nylon deconstruction may be achievable. The best-characterized nylon hydrolases (hereafter, nylonases), $NylC_{p2}$ and NylB, were originally discovered in *Arthrobacter sp.* K172[37]. $NylC_{p2}$, an N-terminal nucleophile (Ntn) hydrolase (EC 3.5.1.117), and NylB (E.C 3.5.1.46), a serine hydrolase with a Ser-Tyr-Lys catalytic triad, are reported to catalyze hydrolysis of oligomers and dimers of PA, respectively (Fig. 1A)[38,39]. Numerous other hydrolases have potential PA depolymerization activity, including Ser-(*cis*)Ser-Lys containing amidases (EC 3.5.1.4), cutinases (EC 3.1.1.74) using the canonical Ser-His-Asp (SHD) catalytic configuration and multiple proteases using either serine or cysteine residues as catalytic nucleophiles (EC 3.4.21 & 3.4.22)[27,40–48]. Taken together, these observations suggest that nylonase activity may be achieved by using multiple different enzyme active-site architectures.

With various potential nylonases described, there is an opportunity to assess enzymes with a variety of catalytic residue architectures for their propensity to deconstruct solid PAs. Previous studies have necessarily used a wide variety of analyses and substrates to detect and quantify biocatalytic nylon deconstruction. For instance, while some studies use proxy measurements to evaluate nylon-active enzymes, such as looking for polymer surface changes via rising-height analysis[43,46], or using indicator compounds to identify the release of amine-containing molecules[49–51], others use soluble substrate analogs to probe enzyme kinetics[52], or look for changes in thickness of self-cast nylon films following enzyme application[53]. However, the large diversity of experimental set-ups and substrates used across such studies, makes a direct comparison of previously described enzymes challenging. Hence, to identify nylonases with the most promising characteristics, it would be advantageous to assess enzymes with self-consistent reaction conditions, substrates, experimental protocols and analysis methods, to ensure equitable comparisons[54–56]. Taking inspiration from the PETase field, a high-throughput depolymerization quantification method that can assess the major soluble released products following enzymatic nylon hydrolysis would be ideal[57–59], in conjunction with rigorous characterization of the PA polymer substrates used[5]. Such analysis techniques could be leveraged to accelerate our understanding of nylonase action and may additionally aid in the discovery and engineering of new nylon-active enzymes.

Here, we examined 40 proposed nylonases for their potential for PA6 depolymerization (Fig. 1B), selecting those with a range of enzyme active-site architectures and a variety of previously described substrate preferences. Within the enzyme set, we also included engineered enzymes with high thermostability, a desirable characteristic to enable polymer deconstruction near the glass transition temperature ($T_g$) of the plastic ($T_g$ PA6 = -50–55 °C)[60], whereby substrate accessibility may be increased[61]. These enzymes were tested for their ability to hydrolyze PA6 films, using a high-throughput nylon depolymerization analysis method based on liquid chromatography with tandem mass spectrometry (LC-MS/MS), which accurately and simultaneously detects and quantifies linear-oligomers of PA6 up to pentamers, and cyclic-oligomers up to trimers in a single sample. Product release analysis was coupled with rigorous characterization of the PA6 materials used in the study, both pre- and post-enzymatic deconstruction. Our results indicate that although measurable nylonase activity is present in multiple enzymes, with different enzyme-dependent distributions of reaction products, Ntn hydrolases appear to be the most active of the enzymes examined here. We present evidence that even the best nylonases tested only hydrolyze a small portion of available PA6, with between 0.1 and 0.7% of the PA6 depolymerized, depending on the enzyme used. The most promising enzyme candidates were

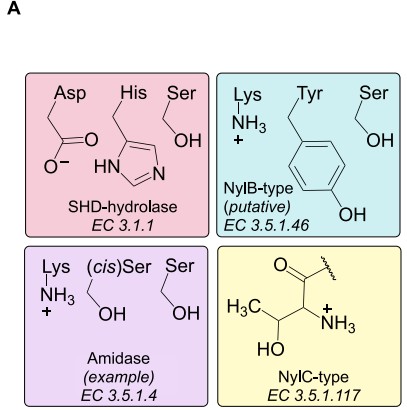

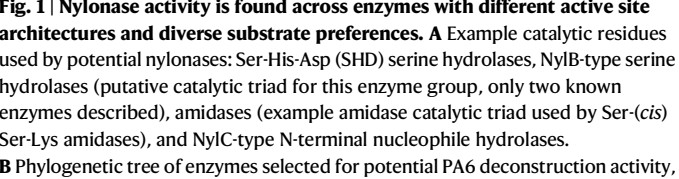

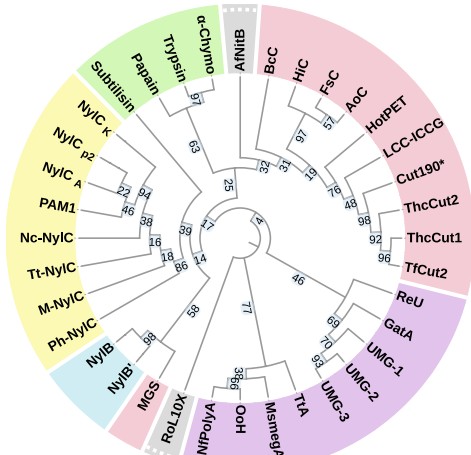

**Fig. 1 | Nylonase activity is found across enzymes with different active site architectures and diverse substrate preferences. A** Example catalytic residues used by potential nylonases: Ser-His-Asp (SHD) serine hydrolases, NylB-type serine hydrolases (putative catalytic triad for this enzyme group, only two known enzymes described), amidases (example amidase catalytic triad used by Ser-(*cis*) Ser-Lys amidases), and NylC-type N-terminal nucleophile hydrolases. **B** Phylogenetic tree of enzymes selected for potential PA6 deconstruction activity,

colored to match the enzyme groupings used throughout (yellow: NylC-type, blue: NylB-type, purple: amidase, pink: SHD-hydrolase, green: protease, gray: misc.). Inset numbers on the branches represent confidence values (%). References for the enzyme source and group are available in Supplementary Data 1, enzyme sequences, EC numbers, and expression conditions are available in Supplementary Data 2. α-Chymo α-Chymotrypsin, HotPET HotPETase, MGS MGS0156, UMG-1 UMG-SP-1, UMG-2 UMG-SP-2, UMG-3 UMG-SP-3.

subjected to additional characterization to explore depolymerization extent as a function of reaction conditions. Because PA6 hydrolysis plateaued in all cases, we further investigated reaction progression with the top-performing enzyme, a thermostabilized NylC homolog, $NylC_K$-TS, to better understand the depolymerization process. For $NylC_K$-TS, no further depolymerization was observed after fresh enzyme addition, but additional product release was observed following addition of fresh PA6, indicating that enzyme failure does not explain low depolymerization extents, and instead low extents of deconstruction may be considered a substrate-based phenomenon. Overall, our study highlights areas for future improvement necessary to create a viable PA6 biocatalytic recycling strategy.

## Results

### Identification, expression, and amide bond hydrolysis assays of potential nylonases

We first constructed a diverse set of enzymes, initially focusing on biocatalysts with reported PA polymer or linear-oligomer deconstruction activity from peer-reviewed literature and patents (Fig. 1B, Supplementary Data 1–2). As polymer chains become more accessible near the glass transition temperature ($T_g$) of the substrate (for PA6, $T_g$ ~ 50–55 °C), enzyme thermotolerance was chosen as a desirable characteristic, and hence, was considered when selecting candidates[62]. As demonstrated by Negoro et al., $NylC_{p2}$ and its homologs (Ntn hydrolases) have previously been shown to exhibit activity on PA6 cyclic oligomers, linear oligomers and solid PA6 (powder and thin film)[24,37,49,53,63], whilst NylB and its homologs have been shown to exhibit PA6 dimer hydrolysis activity[38,39]. Amidases have also been used for PA6 modification. One of these enzymes, NfPolyA, an amidase signature (AS) family protein from *Nocardia farcinica*, had also been reported to exhibit polyurethane (PUR)-hydrolyzing activity, hence, we included some known polyurethanases and urethanases, hypothesizing that these enzymes may also exhibit activity on PA[23,64]. Additional Ntn hydrolases and AS family enzyme candidates were selected following $NylC_{p2}$ and NfPolyA homolog searches, selecting enzymes from potentially thermotolerant sources. Numerous reports suggest serine hydrolases with the canonical Ser-His-Asp (SHD) catalytic triad including cutinases, and proteases using either serine or cystine catalytic nucleophiles, exhibit promiscuous activity for nylon surface modification[44–48,50,65]. Thus, these enzymes were also added to the set. A number of the cutinases described also had PET depolymerization activity, hence, several other PET-active SHD hydrolases were added, including some previously engineered thermostabilized variants[21,58], with the hope that they may also exhibit PA depolymerization.

Following selection, enzymes were broadly categorized into six main groupings dependent on both their catalytic activities and previously described substrate preferences (Fig. 1B, Supplementary Data 1). Enzymes found by homology searches were put in the same group as the query enzyme, whilst those where no natural substrate was described in the original publications were assigned functionality using BLAST searches[66]. The groupings used from here on are as follows: (1) NylC-type enzymes, Ntn-hydrolases found to hydrolyze PA6 trimers and longer PA oligomers, (2) NylB-type enzymes, serine hydrolases (Ser-Tyr-Lys catalytic triad) previously demonstrated to specifically hydrolyze PA dimers, (3) amidases, AS family enzymes that act on linear amides or urethane bond containing compounds, (4) Ser-His-Asp hydrolases (SHD-hydrolases), serine hydrolases with a substrate preference for ester-linked polymers, (5) proteases, serine and cystine proteases that cleave peptide bonds in proteins, and (6) miscellaneous (misc.), enzymes that do not fall into one of the other five main groupings. Manganese peroxidases (MnPs), which have previously been implicated in PA hydrolysis[30,35], were excluded from our study as their radical-based reaction leads to diverse product mixtures that may be less desirable for a closed-loop circular recycling process[6,22].

Inspired by the successful homology-based thermostability enhancement of $NylC_{p2}$ by Negoro et al. ($NylC_{p2}$ D122G/H130Y/D36A/E263Q, $T_m$ increase of 36 °C, demonstrated hydrolytic activity on solid PA6 powder)[49], we also rationally mutated NylCs from *Agromyces* ($NylC_A$) and *Kocuria* ($NylC_K$) to increase their melting temperatures ($T_m$s)[37,49,67]. In both enzymes, residues previously described as potentially destabilizing were substituted for their homologous counterparts in $NylC_{p2}$ (S111G and A137L), with E263Q additionally being introduced into $NylC_A$ as a potential stabilizing mutation from $NylC_K$. These mutations increased protein $T_m$ by 16.4 °C and 25.1 °C for $NylC_K$-TS and $NylC_A$-TS, respectively ($NylC_K$-TS, $T_m$ = 87.4 °C, $NylC_A$-TS, $T_m$ = 87.1 °C, Supplementary Fig. 1). As a three-point mutant of NylB' (NylB homolog from *Arthrobacter sp*. K172, NylB'-SCY: R187S/F264C/D370Y)[39] was demonstrated to have enhanced PA6 dimer hydrolysis activity, we introduced the corresponding mutations into NylB (from *Arthrobacter sp*. K172), to generate NylB-SCY, hoping that the mutations would impart additional PA depolymerization activity. The resulting panel of WT and engineered enzymes comprised of 41 candidates.

Five enzymes were procured from commercial sources; of the 36 enzymes not available commercially, 35 were successfully expressed and purified from *Escherichia coli* (expression summaries detailed in Supplementary Data 2), to give a 40-member candidate pool. To test the ability of each enzyme for amide bond hydrolysis, candidates were incubated with N-(4-nitrophenyl)butanamide (N4NB), a small molecule surrogate that contains a butyramide motif akin to the amide bonds in PA6 (0.5 mM N4NB, 2 μM enzyme, 30 °C in reaction buffer (100 mM sodium phosphate buffer, pH 7.5, 150 mM NaCl), Fig. 2, Supplementary Fig. 2A-I). Successful hydrolytic cleavage of the amide bond in N4NB results in the release of p-nitroaniline (p-NA), which can be monitored spectrophotometrically. The amidases[64,68–71] (Supplementary Fig. 2D–F) and NylB-type enzymes[39,72,73] (Supplementary Fig. 2A) exhibited the highest levels of amide hydrolysis activity, with the UMG-SP enzymes[23] (metagenomic amidases), NfPolyA[64] (polyamidase from *N. farcinica*), GatA[69] (amidase of unspecified origin), and MsmegA (metagenomic amidase) completely converting 500 μM N4NB in under five minutes, whilst ReU[70] (*Rhodococcus equi* TB-60 urethanase), OoH[71] (ω-octalactam hydrolase), and NylB'-SCY[39] reached 100% conversion in under 30 minutes. The NylC-type enzymes[37,67,74,75] (Supplementary Fig. 2B, C) and SHD-hydrolases[21,42,58,76–82] (Supplementary Fig. 2H, I) exhibited a wider range of activities. The highest performers in these two groups, $NylC_A$[67] and AoC[76] (*Aspergillus oryzae* cutinase), displayed 16.2 and 29.3 μM/h p-NA production, respectively, during the linear phase of the reaction, and the poorest performers, M-NylC (*Microbacterium sp*. NylC) and LCC-ICCG[21] (engineered cutinase from leaf/compost metagenome), only had detectable activity after 8 h and 12 h, respectively. Interestingly, none of the proteases (α-chymotrypsin, papain, subtilisin, and trypsin[48], Supplementary Fig. 2G) had any detectable hydrolysis activity under the tested reaction conditions; this may be due to the difference in chemical properties between N4NB and the protease's natural substrates.

### Development of a high-throughput solid PA6 depolymerization analysis method by LC-MS/MS

We next sought to thoroughly characterize the ability of the potential nylonase panel to deconstruct solid PA6. Due to the lack of a strong chromophore in the predicted products and the desire for sub part per million (ppm) detection limits, we developed a high-throughput liquid chromatography with tandem mass spectrometry (LC-MS/MS) analysis method (published on protocols.io, [https://doi.org/10.17504/protocols.io.6qpvr3k92vmk/v1])[83] to monitor the major predicted soluble released compounds following PA6 depolymerization reactions. Utilizing tandem mass spectrometry enabled rapid analyses. When operating in multiple reaction monitoring (MRM) mode, the instrumentation and detection is optimized for each compound,

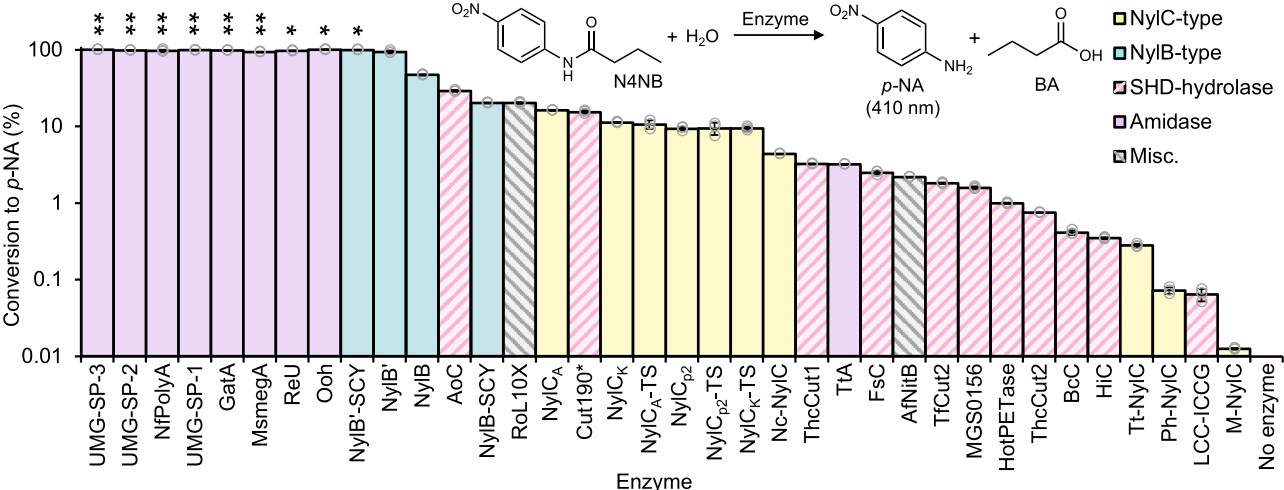

**Fig. 2 | *N*-(4-nitrophenyl)butanamide (N4NB) activity assays with potential nylonases.** All successfully expressed enzymes were tested for their ability to hydrolyze the amide bond in N4NB, leading to the release of *p*-nitroaniline (*p*-NA, monitored spectrophotometrically at 410 nm) and butyric acid (BA). The bar chart shows the conversion extent of N4NB to *p*-NA after 5 h of reaction using 0.5 mM N4NB and 2 μM enzyme at 30 °C in reaction buffer (100 mM sodium phosphate buffer (NaPi), pH 7.5, 150 mM NaCl). Enzymes are colored by their group from Fig. 1, with SHD-hydrolase and misc. group bars hashed to aid clarity; the proteases exhibited no detectable activity and thus are not shown. A double asterisk (**) indicates reactions with enzymes that went to completion in ≤ 5 min, a single

asterisk (*) indicates reactions with enzymes which went to completion in ≤ 30 min. All reactions are ordered with respect to reaction rate, with the fastest enzymes to the left. A no enzyme control reaction led to no detectable release of *p*-NA. Data shown are baseline corrected. Reactions were carried out in triplicate (*n* = 3), error bars show the standard deviation of the replicate measurements, the error bar centers are the means of the replicate measurements, and the replicate measurements are represented as gray circles. References for the enzyme source and group are available in Supplementary Data 1, and the enzyme sequences, EC numbers, and expression conditions are in Supplementary Data 2. Source data are provided in the Source Data file.

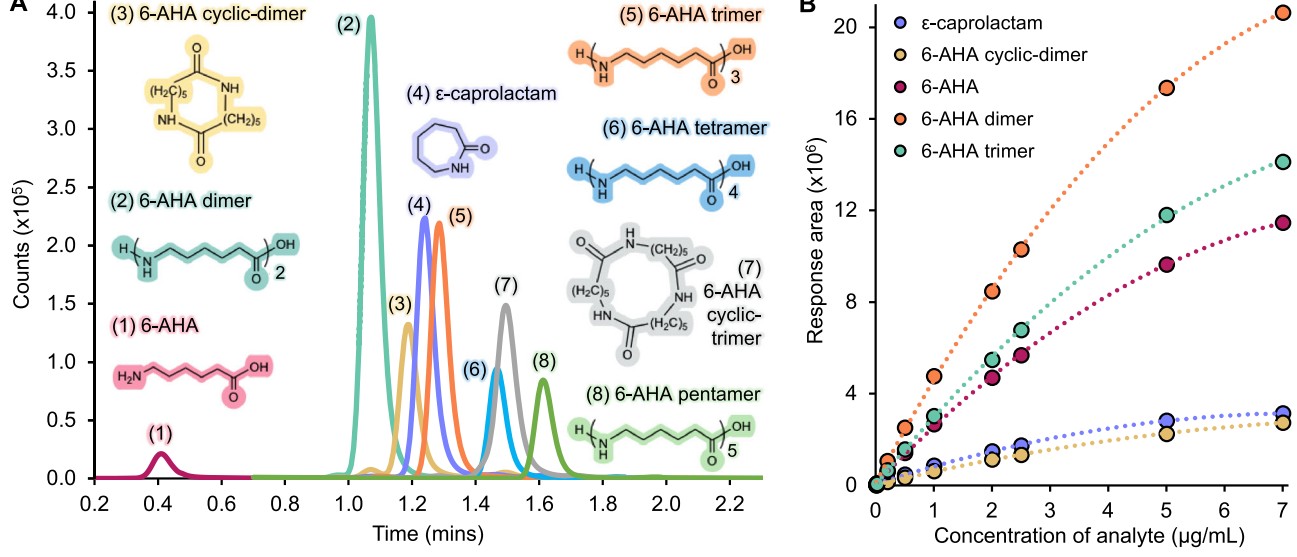

**Fig. 3 | LC-MS/MS analysis of major soluble compounds present in enzymatic PA6 deconstruction reactions. A** Representative LC-MS/MS chromatogram following enzymatic hydrolysis of PA6 (0.01 μM NylC$_K$-TS, 13 mg PA6, 60 °C), showing the eight compounds that were detected and quantified. **B** Representative calibration plot of standards run prior to analysis of samples by LC-MS/MS, showing the five chemical standards (analytes) that were available commercially, measured

from 0.01-7.0 μg/mL. The 6-AHA tetramer and pentamer concentrations, and the 6-AHA cyclic-trimer concentrations were determined using the calibration curves from the 6-AHA trimer and 6-AHA cyclic-dimer, respectively, as no commercial standards were available. The dotted line represents the quadratic line of best fit for each analyte. Source data are provided in the Source Data file.

making the mass transitions highly selective and unique to each analyte, and thus eliminating the need for chromatographic peak separation or product derivatization, significantly reducing analysis time.

Using this ex-situ analysis method, linear-oligomers of 6-aminohexanoic acid (6-AHA) from monomer to pentamer and cyclic-oligomers from ε-caprolactam to trimer were all detected in under three minutes, representing a significant advancement in the speed

and diversity of potential products quantified (Fig. 3A). The use of a multisampler allows eight 96-well plates to be analyzed without manual intervention, with around 20 samples being analyzed per hour. Reaction samples were arrayed across 96-well plates, with a calibration verification standard (CVS) evaluated every 20 wells to ensure accurate quantification and system stability, demonstrating the suitability of this method for high-throughput analyses[83]. All analytes were detectable from 0.01–7.0 μg/mL, with samples above the upper quantitation

limit further diluted to enable quantification in the calibration range. Calibration curves were run prior to each set of samples analyzed to account for mass spectrometry signal drift (Fig. 3B). The linear 6-AHA products from monomer to trimer and cyclic products from ε-caprolactam to 6-AHA cyclic-dimer were quantified using commercial standards. The linear 6-AHA tetramer and pentamer were quantified by applying the calibration curve from 6-AHA linear trimer; the 6-AHA cyclic-trimer was quantified utilizing the calibration curve from the 6-AHA cyclic-dimer responses.

## Assessment of potential nylonases on a solid PA6 substrate

With a robust nylon depolymerization analysis method in hand, we next sought to examine the capabilities of our enzyme panel for PA6 deconstruction. For activity screens, commercially available PA6 film from Goodfellow was used as the substrate (13.2% crystallinity by differential scanning calorimetry (DSC), 0.2 mm thickness, full material characterization detailed in Supplementary Table 1, with DSC, gel permeation chromatography (GPC) and thermogravimetric analysis (TGA) plots shown in Supplementary Fig. 3A–C).

PA6 film was washed with DI water prior to use, as detailed in the Methods section, to remove surface-bound cyclic-oligomer by-products created during the nylon manufacturing process[84]. Reactions comparing the enzyme panel were conducted for 10 days at temperatures from 40 to 70 °C, with time points taken at 24, 72, 168, and 240 h (Supplementary Fig. 4). Control assays without enzyme led to minimal release of linear 6-AHA oligomers, with the 6-AHA monomer below the limit of detection across all tested temperatures when no enzyme was present (Supplementary Fig. 5A). Enzyme reactions were carried out in 100 mM sodium phosphate buffer (NaPi), pH 7.5, 150 mM NaCl, and contained 2 μM enzyme and 13 mg PA6 (two squares of 0.5 × 0.5 cm PA6 film), representing a substrate loading and an enzyme/substrate loading in the reactions of 0.65 wt% and 0.15 mM enzyme/g PA6 film, respectively, unless otherwise stated. Although polymer deconstructions may be surface-area limited processes, the enzyme/substrate loading is referred to throughout as amount of enzyme per mass of PA6 as is good practice for enzymatic depolymerization reactions[54,56,85]. Across all tested enzymes, the extent of ε-caprolactam release was consistent; for most enzymes assayed, concentrations of the 6-AHA cyclic-dimer and cyclic-trimer also remained constant during the reaction (Supplementary Fig. 5B). Hence, for clarity, only the linear products are presented graphically, with cyclic products mentioned only when their levels changed. The linear oligomers (from monomer to trimer) appear robust to incubation for extended time periods at 40–80 °C, with an average recovery (as measured by LC-MS/MS) across all time points and temperatures of 96.4% ± 9.2, 99.9% ± 6.3 and 104.3% ± 7.2 of individually incubated 10 μg/mL standards of 6-AHA monomer, 6-AHA dimer and 6-AHA trimer, respectively (Supplementary Table 2). These results indicate that changes in oligomer levels seen in reactions can be attributed mainly to enzyme action. As a note, although the nylonase deconstruction assays presented below were all conducted with purified proteins, the LC-MS/MS analytical method is unaffected by instead using crude cell lysate (Supplementary Fig. 6) as the biocatalytic agent, with a similar product release profile witnessed as compared to using purified protein (Supplementary Fig. 7). To aid comparisons of total PA6 depolymerization extent in each reaction, the detected soluble released products are presented as their 6-AHA equivalents.

Nylonase activity was observed across a range of enzymes from the different groups tested: overall, 31 enzymes demonstrated measurable activity, while nine enzymes exhibited minimal detectable additional soluble product release above background levels (Fig. 4, Supplementary Figs. 8–13). The temperatures for optimal activity ranged from 40 to 70 °C, with, in general, higher levels of soluble product release seen with increasing temperatures. Interestingly, there appears to be different distributions of products released across the

different enzyme groups. Namely, NylB-type enzymes, GatA, and UMG-SP-1 mainly produce 6-AHA monomer, whilst most amidases and cutinases release a mixture of linear products, with very little 6-AHA pentamer seen for many of the amidases. The most active NylC-type enzyme reactions, however, are dominated by 6-AHA dimer. It is important to note that only one pH and buffer condition was tested, so it is possible that relationships between different enzyme activities seen here may be altered by more extensive optimization of reaction conditions for each individual enzyme.

By far the most active enzymes were from the NylC-type group, with $NylC_{p2}$, $NylC_A$, $NylC_K$, and their engineered variants identified as the top-performers, being between two and six-fold more active than the best enzymes from other groups. The significant accumulation of linear 6-AHA trimer in the reactions with the NylC variants found via homology searches is probably a result of their lower propensity for trimer hydrolysis. For instance, $NylC_K$-TS hydrolyzes 100% of 50 μM 6-AHA trimer to dimer and monomer in under 30 minutes at reaction conditions, while trimer hydrolysis is much slower with Tt-NylC (*Thermocatellispora tengchongensis* NylC), where 67.1% trimer remains after five hours (pH 7.5 NaPi buffer, 150 mM NaCl, 2 μM enzyme, 60 °C) (Supplementary Fig. 14A, B). As the 6-AHA dimer remains intact at reaction conditions in the presence of $NylC_K$-TS after 24 h (100 μM dimer, pH 7.5 NaPi buffer, 150 mM NaCl, 2 μM $NylC_K$-TS, Supplementary Fig. 14C), monomer accumulation in reactions containing $NylC_{p2}$, $NylC_A$, $NylC_K$, and their variants may be due to the fast hydrolysis of released trimers to 6-AHA dimer and monomer. Thermostabilizing mutations appeared beneficial as hypothesized, with all NylC-TS variants showing increased activity at 70 °C (Supplementary Fig. 8, Fig. 4), compared to 50-60 °C optimal temperatures for their wild-type (WT) counterparts. However, interestingly, these mutations also appeared to slow the hydrolysis of the 6-AHA cyclic-trimer, which is a known substrate for NylC-type enzymes (Supplementary Fig. 15)[74]. The best NylC-type enzyme tested was the two-point mutant $NylC_K$-TS ($T_m$ = 87.4 °C) (Figs. 4 and 5A), which was even active up to 80 °C (Supplementary Fig. 16). In comparative reactions of $NylC_K$-WT and $NylC_K$-TS at 40-60 °C the total released products and reaction profiles are highly comparable (1 μM enzyme and 13 mg PA6, 0.08 μM/g PA6, 0.65 wt% substrate loading, pH 7.5 NaPi buffer, 150 mM NaCl, Supplementary Fig. 16). However, from 70-80 °C, $NylC_K$-TS has a higher overall yield of soluble products after 10 days (Supplementary Fig. 16). Comparing both enzymes at the temperatures which led to the highest product release (50 °C and 80 °C, respectively), $NylC_K$-TS starts producing more 6-AHA equivalents from 3 h of reaction onwards, releasing 28.8% more 6-AHA equivalents over the course of 10 days (1 μM enzyme and 13 mg PA6, 0.08 mM enzyme/g PA6, 0.65 wt% substrate loading, pH 7.5 NaPi buffer, 150 mM NaCl, Fig. 5B). Despite the promise of the overall best performing enzyme $NylC_K$-TS, the highest soluble PA6 oligomer release seen (339.7 μM of 6-AHA equivalents) only equates to 0.67 wt% PA6 depolymerization extent.

Intriguingly, although NylB appears sparingly active for PA6 film deconstruction, NylB', a NylB homolog with 88% sequence identity, released 72.9 μM 6-AHA equivalents over 10 days at 40 °C (Fig. 4B, Supplementary Fig. 9). The three-point NylB' variant, NylB'-SCY, is even more active and additionally more thermostable, with optimal 6-AHA equivalent release at 50 °C (130.2 μM 6-AHA equivalents), corresponding to 0.26 wt% PA6 deconstruction after 10 days of reaction. Although the NylB'-WT scaffold appears to be the driving force for increased PA6 hydrolysis activity compared to NylB-WT, the R187S/F264C/D370Y mutations also appear to have an effect as NylB-SCY is three-fold more active than NylB-WT. However, as the total 6-AHA equivalents released over 10 days at 40 °C by NylB-SCY was <8.0 μM, the overall effect on PA6 depolymerization was negligible. NylB'-SCY readily hydrolyzes both 6-AHA dimer and trimer at reaction conditions (pH 7.5 NaPi buffer, 150 mM NaCl, 2 μM NylB'-SCY, 50 °C), with 100 μM dimer or 50 μM trimer completely converted to monomer in under

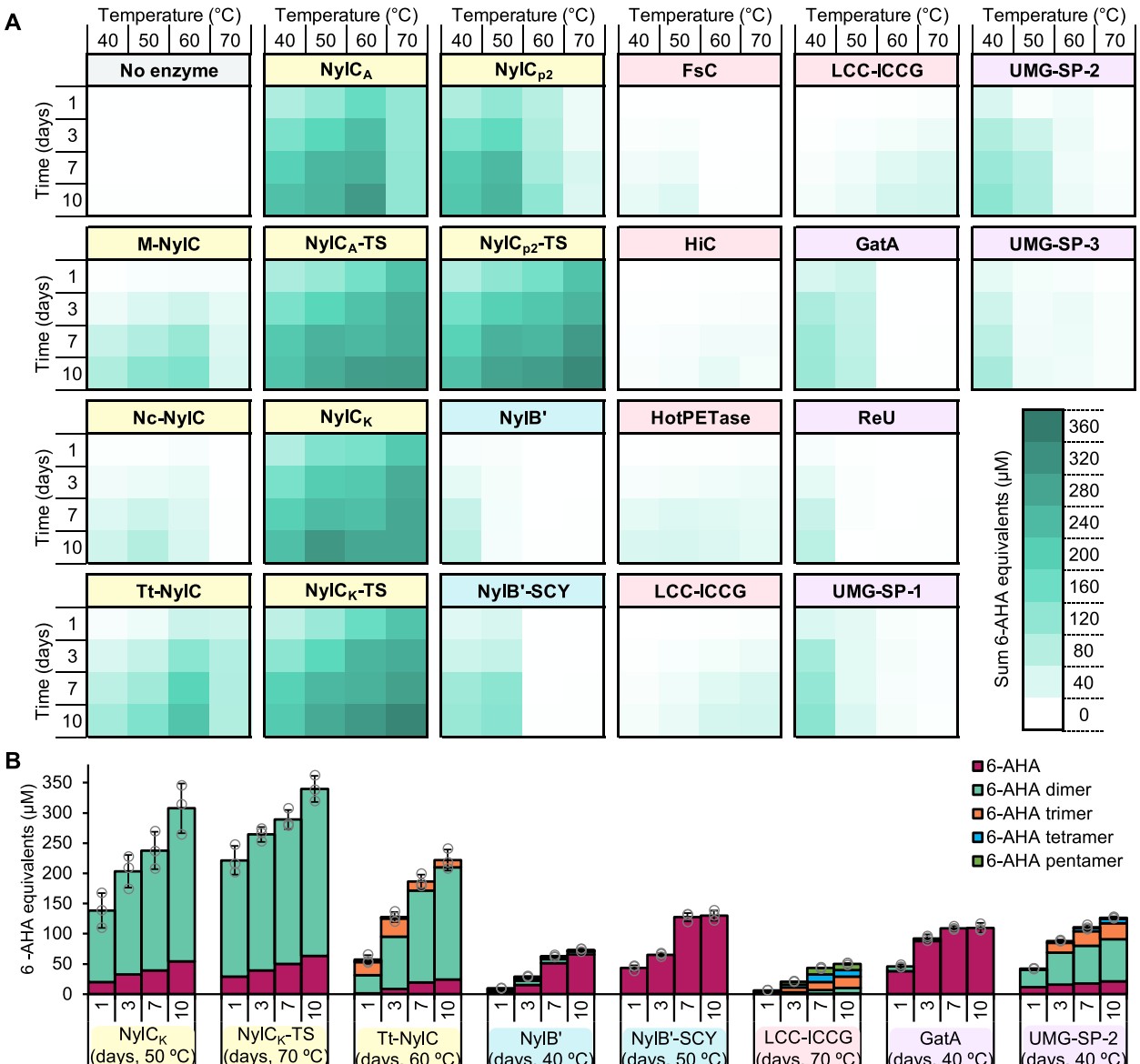

**Fig. 4 | Enzymatic PA6 depolymerization reactions with the top performing enzymes from each group. A** Heat map time-course profiles of the total released (sum) 6-AHA monomer equivalents following PA6 depolymerization reactions with the top performing NylC-type (yellow), NylB-type (blue), SHD-hydrolase (pink), and amidase (purple) enzymes, carried out from 40 to 70 °C over 10 days. The heat map gradient represents the released 6-AHA monomer equivalents from 0 to 360 µM, with each square representing the average of reactions carried out in triplicate (n = 3). (Full enzyme names and sources are in Supplementary Data 1.) **B** Representative depolymerization reactions with enzymes from each group at their

optimal temperatures (reactions conducted between 40 and 70 °C, x-axis), with 6-AHA oligomers of different lengths represented as their 6-AHA monomer equivalents. Reactions were carried out in triplicate (n = 3), error bars show the standard deviation of the replicate measurements, the error bar centers are the means of the replicate measurements, and the replicate measurements are represented as gray circles. For all data in (**A**) and (**B**), the reactions contained 2 µM enzyme and 13 mg PA6 (0.15 mM enzyme/g PA6, 0.65 wt% substrate loading) and were incubated in reaction buffer (100 mM NaPi buffer, pH 7.5, 150 mM NaCl). Source data are provided in the Source Data file.

30 minutes (Supplementary Fig. 17A, B). Based on these experimental results, it is difficult to identify whether the accumulation of 6-AHA monomer is a product of longer oligomer release followed by fast hydrolysis to monomer, or whether NylB′-SCY exclusively releases 6-AHA monomer from polymer chain ends.

For the nylon-active SHD-hydrolases, *Fusarium solani* cutinase (FsC) and *Thermobifida cellulosilytica* cutinase 1 (ThcCut1) have been used previously to modify the surface of PA-6,6 fabric[42,47,50]. The reaction profiles and product distributions of all of the biocatalysts tested from this group suggest a similar mode of action, with the mixture of soluble oligomers released indicating non-specific cleavage of surface residues, rather than coordinated, progressive depolymerization activity (Supplementary Fig. 10). Support for this mode of

action comes from the most active SHD-hydrolase tested, LCC-ICCG (Fig. 4B), which produces only 49.8 µM of 6-AHA equivalents over the course of 10 days at 70 °C, corresponding to 0.09 wt% depolymerization of the PA6 film. The observation that mixtures of oligomers are retained over the 10 days of reaction additionally suggests that there is limited further deconstruction of these soluble linear products by SHD-hydrolases. Indeed, incubation of LCC-ICCG with 50 µM 6-AHA trimer for 24 h at reaction conditions (pH 7.5 NaPi buffer, 150 mM NaCl, 2 µM LCC-ICCG, 70 °C), leads to negligible turnover to dimer or monomer (Supplementary Fig. 17C).

Similarly, low extents of depolymerization are also observed for the amidases, potentially due to their low thermotolerance, with most becoming deactivated above 40 °C (Fig. 4A, Supplementary Fig. 11).

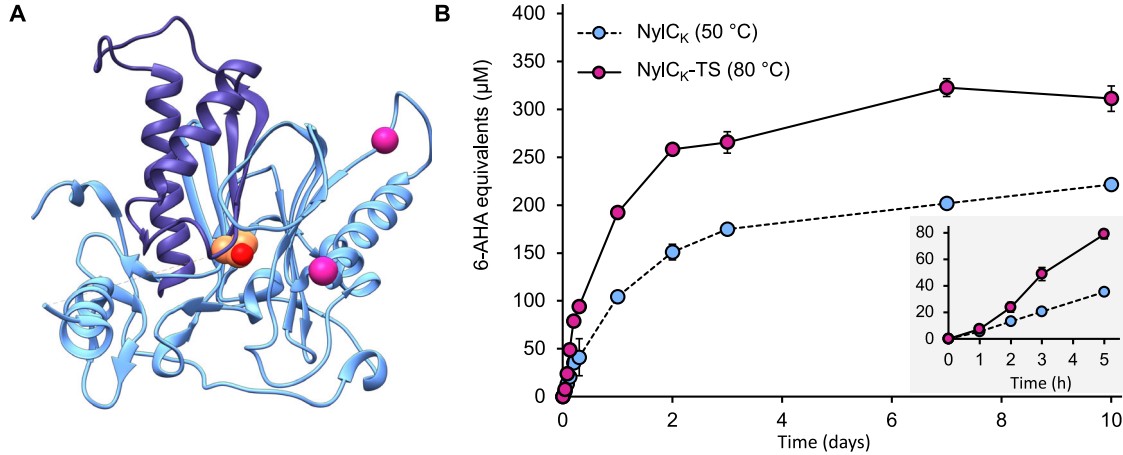

**Fig. 5 | Comparison of NylC$_K$ and the thermostabilized variant NylC$_K$-TS.**
**A** Mutational map of NylC$_K$-TS (NylC$_K$-S111G/A137L) using the crystal structure of the close homolog NylC$_A$ (PDB: 3AXG [https://doi.org/10.2210/pdb3AXG/pdb]) as a scaffold. NylC$_K$-TS is an αβ heterodimer, which is represented by the light and dark blue ribbons; the catalytic residue (T267) is shown as a van der Waals surface colored by all atoms with orange carbon atoms; the thermostabilizing mutations are represented as pink spheres. Residues 261-266 are unmodelled in the PDB structure and are represented as a dashed line. **B** Total released linear 6-AHA

oligomers following PA6 film depolymerization reactions with either 1 μM NylC$_K$ or 1 μM NylC$_K$-TS and 13 mg PA6 (0.08 mM enzyme/g PA6, 0.65 wt% substrate loading), at the temperature where the highest depolymerization extent was seen (reactions conducted from 40 to 80 °C). The gray inset graph is an expansion of the early time points taken during the reaction. Reactions were carried out in triplicate (n = 3), error bars represent the standard deviation of the replicate measurements, and the error bar centers are the mean of the replicate measurements. Source data are provided in the Source Data file.

The limited activity of these enzymes was surprising, as the amidases were among the most active enzymes from the initial screens on N4NB. However, as the active sites of NfPolyA and its homologs are buried within the enzyme core, their ability to bind and access amide bonds in polymer chains is likely more restricted (Supplementary Fig. 18). GatA and UMG-SP-2 produced the most 6-AHA equivalents (109.6 μM and 126.3 μM, respectively, over 10 days at 40 °C) amongst the amidases (Fig. 4B), corresponding to 0.2–0.25% PA6 film depolymerization. The interesting preference of GatA for 6-AHA monomer release is in part due to its nylon oligomer hydrolysis ability; GatA can hydrolyze linear products, with 67% of 100 μM 6-AHA dimer and 100% of 50 μM 6-AHA trimer hydrolyzed under reaction conditions (pH 7.5 NaPi buffer, 150 mM NaCl, 2 μM GatA, 40 °C) within 24 h (Supplementary Fig. 19A, B), but it also hydrolyzes cyclic oligomers. GatA completely depletes 6-AHA cyclic-trimer during PA6 film reactions (Supplementary Fig. 20), with 6-AHA cyclic-dimer also being a substrate for the enzyme (Supplementary Fig. 21A, B), indicating that a proportion of the monomer release observed is the product of hydrolysis of these surface contaminants. Interestingly, UMG-SP-1 also has a preference for monomer release (Supplementary Fig. 11). UMG-SP-1 is similarly able to hydrolyze small PA6 oligomers, hydrolyzing both 100 μM 6-AHA dimer and 50 μM 6-AHA trimer to produce a majority 6-AHA monomer product at reaction conditions (pH 7.5 NaPi buffer, 150 mM NaCl, 2 μM UMG-SP-1, 40 °C) within 24 h (Supplementary Fig. 18C, D), with 6-AHA cyclic-trimer also appearing to be a substrate for this enzyme (Supplementary Fig. 20). Hence, product preference for the amidases does not appear as consistent as with other enzyme groups tested here, with soluble oligomer distributions following PA6 depolymerizations being highly enzyme dependent. As with the SHD-hydrolases, the low rates of depolymerization for all the amidases again currently suggests a surface modification process rather than significant bulk PA6 depolymerization.

No measurable PA6 depolymerization activity was seen for any of the proteases or miscellaneous category enzymes tested here (Supplementary Figs. 12 and 13). A possibility is that these enzymes may cleave polyamides, but release products of a higher molecular weight than what our analysis method detects. However, a lack of subsequent depolymerization down to smaller molecular weight oligomers or monomers suggests that the usefulness of these six enzymes may be limited.

## In-depth characterization of a selection of the best performing enzymes

We were interested to examine a subset of the best performing enzymes, including NylC$_K$-TS, Tt-NylC, and NylB'-SCY, to test if the rate and extent of PA6 depolymerization could be improved and the mode of action understood. NylB'-SCY reactions were carried out at 50 °C, its highest operating temperature, while NylC-type enzyme reactions were conducted at 60 °C to allow fair comparison between homologs. For all three enzymes, increasing the substrate loading from 0.32-1.6 wt% (one to five squares of PA6 film, keeping enzyme concentration constant at 1 μM enzyme, 100 mM NaPi buffer, pH 7.5, 150 mM NaCl) leads to greater product release, most likely due to an increase in available reactive surface area (Fig. 6A). Product distributions remain constant for both NylC$_K$-TS and NylB'-SCY at all substrate loadings, while the 6-AHA trimer accumulates in Tt-NylC depolymerization reactions.

Increasing enzyme loading (keeping PA6 film mass constant at 13 mg PA6 film, 0.65 wt% substrate loading, 100 mM NaPi buffer, pH 7.5, 150 mM NaCl), surprisingly, had little effect on total PA6 film depolymerization (Fig. 6B). 6-AHA equivalent release never surpassed 300 μM under any condition, with improvements in total NylC$_K$-TS depolymerization stalling above 0.1 μM enzyme in the reaction (8 μM enzyme/g PA6). However, interestingly at low NylC$_K$-TS enzyme loadings, where the reaction rate will be slowed, a higher proportion of 6-AHA trimer to pentamer products was observed. A proportion of NylC$_K$-TS PA6 hydrolysis events may therefore be random, producing mixed length oligomers that are subsequently hydrolyzed to dimer and monomer. Slowed oligomer hydrolysis in low enzyme loading reactions additionally suggests a preference of NylC$_K$-TS for release of product from the polymer surface over hydrolysis of oligomers in solution. Reactions with Tt-NylC support this hypothesis as trimer levels were only reduced at very high enzyme loadings. These trends, however, are not observed for NylB'-SCY: even at the lowest enzyme loading, only 6-AHA is produced. As an important consideration for further application, significantly less NylC$_K$-TS was required to achieve the highest levels of total depolymerization (0.1 μM NylC$_K$-TS, 8 μM enzyme/g PA6, equivalent to 0.58 mg enzyme/g PA6), compared to both Tt-NylC or NylB'-SCY (10 μM enzyme, 0.77 mM enzyme/g PA6, equivalent to 54.6 mg enzyme/g PA6 and 66.9 mg enzyme g/ PA6, for Tt-NylC and NylB'-SCY, respectively).

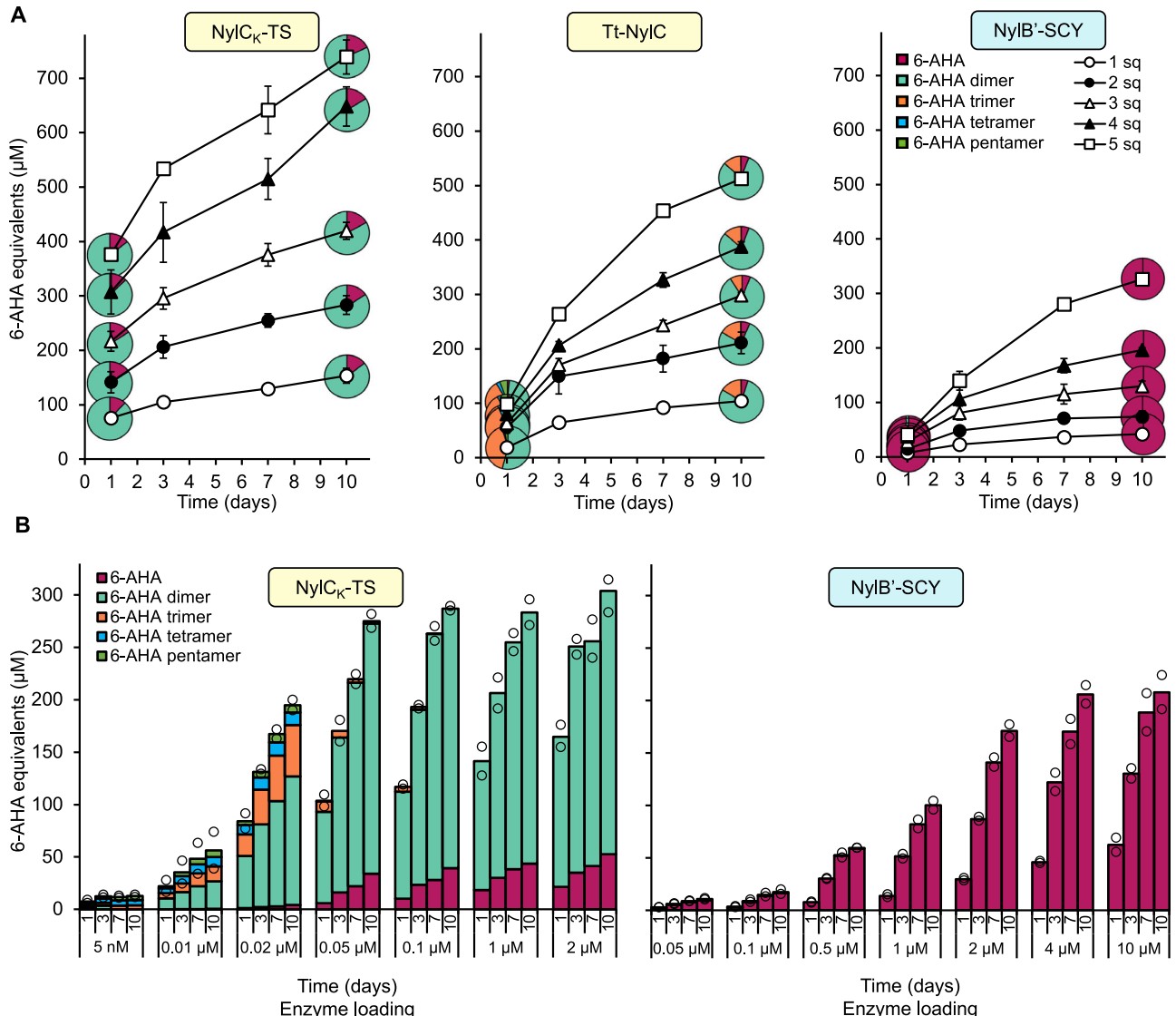

**Fig. 6 | Exploration of reaction conditions for enzymatic PA6 depolymerization. A** Total released linear 6-AHA oligomers, represented as 6-AHA monomer equivalents, in reactions with 1 μM NylC$_K$-TS (60 °C), Tt-NylC (60 °C), or NylB'-SCY (50 °C), over the course of 10 days, varying the substrate loading from 1 to 5 PA6 squares (6.5–32.5 mg PA6, substrate loading of 0.32-1.6 wt%, sq=squares). Pie-charts represent the proportion of different 6-AHA oligomers measured at that time point. Reactions were carried out in triplicate (*n* = 3), error bars show the standard deviation of the replicate measurements, the error bar centers are the means of the replicate measurements, and circle, square, and triangle points represent the mean value of the triplicate measurements. **B** PA6 depolymerization reactions with varying enzyme loadings of NylC$_K$-TS (60 °C), Tt-NylC (60 °C), or NylB'-SCY (50 °C). Reactions contained 13 mg PA6 substrate (0.65 wt% substrate loading) and were monitored for 10 days. Reactions were carried out in duplicate (*n* = 2), and the replicate measurements are represented as gray circles. For all data in panels A and B, the reaction buffer consisted of 100 mM NaPi buffer, pH 7.5, 150 mM NaCl. Source data are provided in the Source Data file.

Variation of reaction pH from pH 6-10 (1 μM enzyme and 13 mg PA6, 0.08 mM enzyme/g PA6, substrate loading 0.65 wt%, 100 mM buffer, 150 mM NaCl) also did not elicit higher levels of PA depolymerization; however, it did reveal differences in pH tolerance amongst the three enzymes (Supplementary Fig. 22). NylC$_K$-TS was mostly unaffected by pH changes, with Tt-NylC being similarly pH-robust with drops in activity only seen at the extremities of the pH range tested. Conversely, NylB'-SCY is particularly sensitive to pH changes: pH 7 is optimal with significant decreases in activity either side of this, and complete deactivation from pH 9-10.

**Examination of the reaction profile of NylC$_K$-TS**
We next sought to identify what was limiting the enzymatic depolymerization of PA6 and causing the observed asymptotic reaction profiles, using the most active enzyme, NylC$_K$-TS, as the test case. We identified slight crystallinity increases in the PA6 substrate in no

enzyme control reactions in reaction buffer (100 mM NaPi buffer, pH 7.5, 150 mM NaCl) at 70 °C (maximum 3.2% increase in crystallinity by DSC over 10 days), hence, reactions were carried out at 60 °C where this effect was less noticeable (maximum 0.2% increase in crystallinity by DSC over 10 days) (Supplementary Table 1, Supplementary Fig. 23). Reactions contained 1 μM of enzyme and 13 mg of PA6 (0.08 mM enzyme/g PA6, substrate loading of 0.65 wt%), in 100 mM pH 7.5 NaPi buffer and 150 mM NaCl. To rule out effects of incubation-induced PA6 changes during the reaction, we conducted NylC$_K$-TS depolymerizations using PA6 film incubated in reaction buffer at 60 °C for either three or seven days prior to enzyme addition. As there was no significant difference between the product release of pre-incubated PA6 film versus non-incubated PA6, this effect could be discounted (Supplementary Fig. 24A).

Interestingly, including bovine serum albumin (BSA) in reactions allows for the same level of PA6 depolymerization with 100-fold less

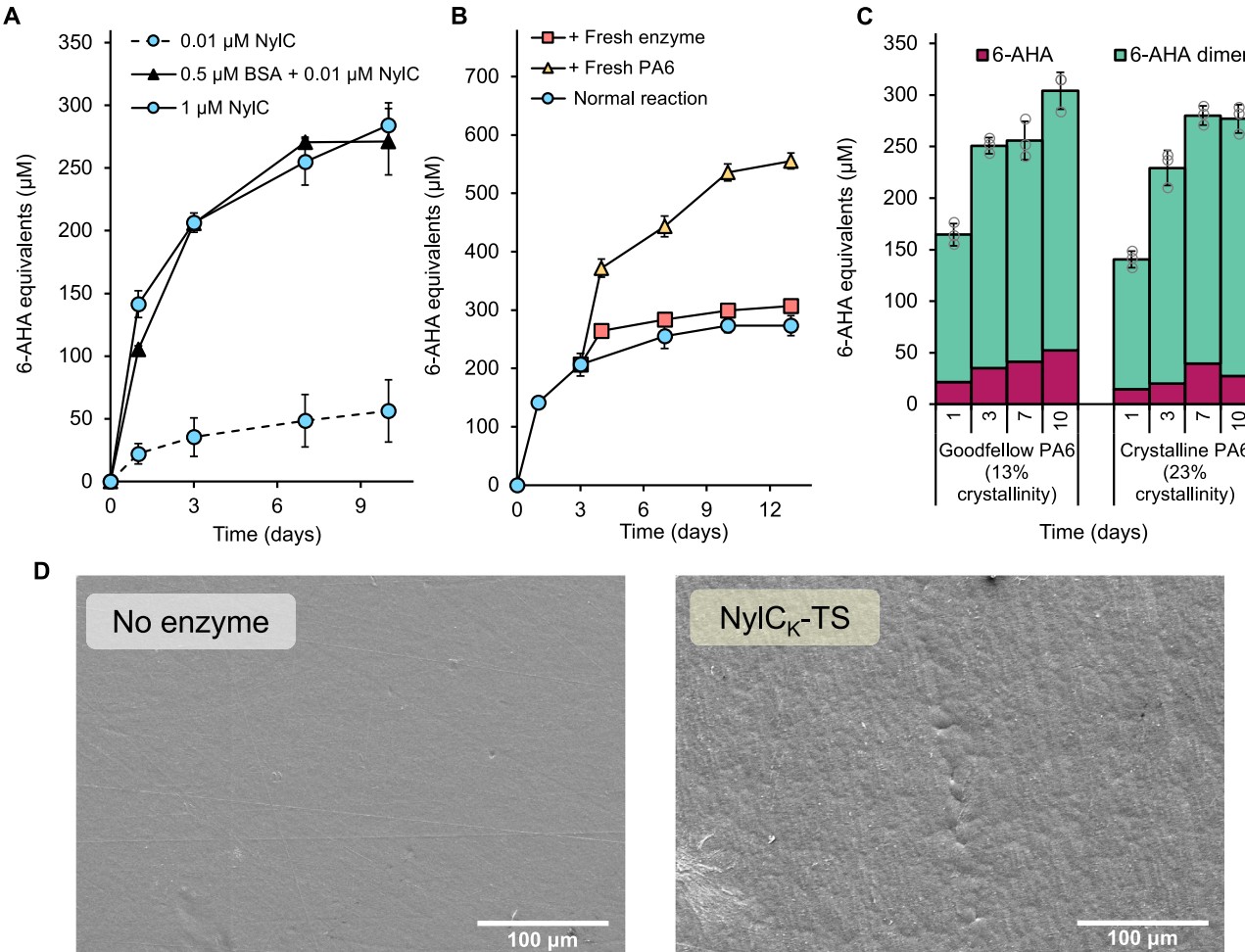

**Fig. 7 | Exploration of PA6 depolymerization plateau using NylC$_K$-TS as a test case. A** Total released linear 6-AHA oligomers following PA6 depolymerization reactions (13 mg PA6, 0.65 wt% substrate loading), using either 0.01 μM or 1 μM NylC$_K$-TS enzyme alone (0.8 μM enzyme/g PA6 or 0.08 mM enzyme/g PA6, respectively) compared to a reaction containing 0.01 μM NylC$_K$-TS supplemented with 0.5 μM bovine serum albumin (BSA) over the course of 10 days. **B** Total released linear 6-AHA oligomers in reactions with 1 μM NylC$_K$-TS and 13 mg PA6 (0.08 mM enzyme/g PA6, 0.65 wt% substrate loading), where the reaction was either allowed to progress for 13 days with no interruption (Normal reaction), where fresh enzyme was added to the reaction after three days (1 μM NylC$_K$-TS, + Fresh enzyme), or where fresh substrate was added to the reaction after 3 days (13 mg PA6, + Fresh substrate), and the reactions monitored for a further 10 days. For (**A**) and (**B**), circle, square, and triangle points represent the mean value of the replicate measurements, error bars represent the standard deviation of the replicate measurements, and the error bar centers are the means of the replicate measurements. **C** PA6 depolymerization reactions with 1 μM NylC$_K$-TS and 13 mg PA6 (0.08 mM enzyme/g PA6, 0.65 wt% substrate loading), with either low crystallinity Goodfellow film (13.2% crystallinity by DSC), or high crystallinity PA6 film (23.0% crystallinity by DSC). Replicate measurements are represented as gray circles. For (**A**)–(**C**), reactions were carried out in triplicate (*n* = 3), error bars represent the standard deviation of the replicate measurements, and the error bar centers are the means of the replicate measurements. **D** SEM images of PA6 films (13 mg) incubated either without enzyme (no enzyme) or with 1 μM NylC$_K$-TS (0.08 mM enzyme/g PA6, 0.65 wt% substrate loading). Additional images are shown in Supplementary Fig. 27. Scale bar, 100 μm. For (**A**)–(**D**), all reactions were carried out in reaction buffer: 100 mM sodium phosphate buffer (NaPi), pH 7.5, 150 mM NaCl at 60 °C. Source data are provided in the Source Data file.

enzyme loading (0.01 μM NylC$_K$-TS, 0.8 μM enzyme/g PA6, substrate loading of 0.65 wt%, 0.5 μM BSA, 100 mM NaPi buffer, 150 mM NaCl, pH 7.5, 60 °C, Fig. 7A, Supplementary Fig. 24B). Increasing the concentration of BSA in PA6 deconstruction assays (from 0.5 to 2 μM), does not promote additional depolymerization, indicating that BSA is not promoting further PA6 hydrolysis, but is instead acting to enhance NylC$_K$-TS specific activity (Supplementary Fig. 24B). Additional evidence for this is that the promotion of activity only occurs at lower NylC$_K$-TS concentrations (0.01 μM enzyme), with addition of BSA to reactions containing 1 μM NylC$_K$-TS leading to no additional benefit (Supplementary Fig. 24C). As BSA is known to prevent non-specific binding in reactions, we believe these findings suggest that the activity of NylC$_K$-TS may be substantially impacted by non-productive PA6 binding events. Alternatively, BSA may coat the reaction vessel, reducing enzyme adsorption to these surfaces, hence promoting mass

transfer of NylC$_K$-TS to the polymer surface. More extensive exploration of this phenomenon may be necessary to help guide future NylC$_K$-TS engineering efforts or may suggest new ways to promote faster or additional PA6 depolymerization.

Reaction progress was not recovered by supplementation with fresh NylC$_K$-TS after either three or seven days of enzyme reaction (Fig. 7B, Supplementary Fig. 24D). However, addition of new PA6 substrate (13 mg) to a NylC$_K$-TS depolymerization reaction that had been running for three days (the start of reaction plateau), led to additional product release of a similar magnitude as from the original substrate, with the same relationship seen after substrate addition to a seven-day reaction (Fig. 7B, Supplementary Fig. 24D). Reaction profiles following new substrate addition match 6-AHA equivalent release seen in standard reactions like those shown in Figs. 5 and 6, indicating that NylC$_K$-TS retains almost full activity after both three and seven days of

reaction at 60 °C, and that there are no inhibitory compounds present which could explain reaction stalling. Characterization of the PA6 films by GPC suggests that there were no extensive changes in the number average molar mass ($M_n$) or molar mass dispersity ($Đ$) of the PA6 polymer chains following 10 days of enzyme incubation with NylC$_K$-TS at any temperature (Supplementary Table. 3, Supplementary Fig. 25A). There were also no significant differences in the percentage crystallinity of the PA6 substrate incubated with or without enzyme over the course of 10 days, as measured by DSC (Supplementary Fig. 25B). As a note, TGA analysis revealed that the PA6 film absorbed an average of 1.9 wt% water following 10 days incubation in reaction buffer across all temperatures (no enzyme controls), and there was no noticeable difference in substrate water absorption in reactions with NylC$_K$-TS (average of 1.7 wt% across all temperatures) (Supplementary Table. 3, Supplementary Fig. 25C).

Taken together, these results suggest that the reaction plateau for NylC$_K$-TS is a consequence of lack of remaining hydrolysable substrate for this enzyme following 10 days of reaction. Furthermore, typically for more extensive enzymatic plastic depolymerizations, both rate and extent of deconstruction are highly sensitive to substrate crystallinity[61]. However, for PA6 depolymerization with NylC$_K$-TS, using a more crystalline PA6 film substrate (23.0% crystallinity by DSC, prepared in house, full material characterization in Supplementary Table 1, Supplementary Fig. 26), leads to a comparable amount of 6-AHA equivalents release compared to reactions with the Goodfellow film that is more amorphous (13.2% crystallinity by DSC, Fig. 7C). Hence, we can conclude that NylC$_K$-TS is likely only working on a small amount of accessible nylon polymer on the film surface, and so does not reach the depolymerization extents at which substrate crystallinity would play a role in reaction progression. Indeed, SEM images of PA6 incubated with NylC$_K$-TS (1 μM enzyme, 0.08 mM enzyme/g PA6, 0.65 wt% substrate loading, 100 mM pH 7.5 NaPi buffer, 150 mM NaCl, 60 °C, 10 days), reveal a slight surface roughening, but no significant pits or features that are commonly associated with more extensive biocatalytic polymer deconstruction (Fig. 7D, Supplementary Fig. 27)[58,86]. As these results indicate a surface area-limited process, we attempted to alter the surface area of the available PA6 substrate whilst retaining the same mass. Typically, this is done by using a powdered substrate; however, we found that enzymatic assays with PA6 powder (sourced from Goodfellow, particle size: 5–50 μm, full material characterization detailed in Supplementary Table 3) were very difficult to standardize due to reaction volume changes over the course of incubation (Supplementary Fig. 28). Additionally, the powder exhibited a high crystallinity (47.6% crystallinity by DSC), which may be undesirable. Hence, we conducted assays using the same mass of a thicker PA6 film (PA6 thick film, 0.5 mm thickness, 13 mg = two 0.32 ×0.32 cm squares, sourced from Goodfellow, full material characterization detailed in Supplementary Table 3), representing a ~ 50% reduction in PA6 surface area in each reaction compared to our standard 0.2 mm thickness film. Using the PA6 thick film, product release proceeded at a slower rate, with 34% fewer 6-AHA oligomers released over the course of 10 days, compared to the standard PA6 film (1 μM of enzyme, 0.08 mM enzyme/g PA6, 0.65 wt% substrate loading, 100 mM pH 7.5 NaPi buffer, 150 mM NaCl, 60 °C, Supplementary Fig. 29), confirming that surface area is indeed a factor controlling both rate of reaction and extent of depolymerization.

## Discussion

There is increasing interest in expanding the successes of enzymatic PET deconstruction to other plastics including nylon[21,57,58,87,88]. To effectively accomplish this requires both candidate enzymes to underpin the discovery, engineering, and evolution of improved enzymes, and suitable analytical methods. Using a high-throughput LC-MS/MS-based analysis strategy, we explored the PA6 deconstruction capacity of a diverse panel of potential nylonases, including candidates not previously examined for PA6 hydrolysis. We demonstrate that enzymes from a wide variety of sources exhibit sparing abilities to release soluble oligomeric products from a PA6 film. Enzymes with the canonical Ser-His-Asp catalytic triad and the amidases were shown to exhibit low propensities for PA6 depolymerization. Perhaps unsurprisingly, there was little correlation between the capacity for an enzyme to hydrolyze the soluble N4NB substrate relative to solid PA6 deconstruction: our findings indicate that soluble substrates such as N4NB may be used to indicate poor nylonase activity but are not good indicators of high nylonase performance on solid PA6. Features necessary for plastic depolymerization, such as enzyme-polymer surface binding and release, ability to hydrolyze an insoluble solid, and potential for the active site to accommodate bulky polymer chains, tend not to be well captured by soluble substrate hydrolysis assays, hence, studies of this kind, which accurately identify potential nylonases on realistic substrates, are vital. Although it is unlikely that the SHD-hydrolases and amidases studied here are immediately suitable for a PA6 recycling process, for reasons ranging from low depolymerization rates, thermal stability, poor expression levels, and mixed pools of reaction products[89,90], they may find use in fabric industries. Cutinases and amidases have been previously used to improve the surface qualities of nylon fabric, whilst GatA, an enzyme initially identified as a polyurethanase, may prove useful for improving the quality of spun fibers, by hydrolyzing undesirable cyclic nylon polymerization by-products[47,50]. Interestingly, many of the nylon-active cutinases have previously been implicated in PET hydrolysis[21,58,91–93], with a number also having demonstrated polyurethanase and urethanase activity[23,64,69,70,82,94], indicating that the cross-reactivity of enzymes may be something that can be exploited to discover biocatalysts for different plastic deconstruction campaigns in the future.

Although NylB-type enzymes have previously reported activity on various soluble nylon oligomers, we demonstrate that they can act also on solid PA6, with NylB'-SCY being the most active (highest depolymerization extent seen of 0.42 wt% PA6 film using 10 μM NylB'-SCY, 0.77 mM enzyme/g PA6, equivalent to 66.9 mg enzyme/g PA6). The wide difference in activity between NylB and NylB' for PA6 film hydrolysis was particularly surprising, as NylB is 200 times more active for 6-AHA dimer hydrolysis than NylB'[39]. With 6-AHA monomer being the sole product under all conditions, it may be that NylB'-SCY is acting in exo-type manner, cleaving terminal monomer moieties from the PA6 film surface, or that the hydrolysis of released longer oligomers is too fast to observe. In either case, the release of a homogenous reaction product may be desirable for downstream applications[18], such as cyclization of 6-AHA to form caprolactam for new nylon production[95,96].

The rationally mutated NylC$_K$-TS variant exhibited the highest extents of depolymerization, indicating that thermostability is indeed a useful feature for PA6 deconstruction. Introduction of just two mutations led to an increase in $T_m$ of 16.4 °C (NylC$_K$-TS, $T_m$ = 87.1 °C), indicating that the enzyme is highly engineerable. NylC$_K$-TS proved very stable, being active in reactions up to 80 °C, and retaining activity after even seven days of reaction at 60 °C, and it also required the lowest enzyme loadings to achieve the highest levels of PA6 depolymerization (0.1 μM NylC$_K$-TS, 8 μM enzyme/g PA6, equivalent to 0.58 mg enzyme/g PA6), both desirable characteristics of an industrial biocatalyst. Our analyses indicate that NylC$_K$-TS appears to cleave the polymer in a non-specific manner, with a mixture of soluble oligomers initially released before further hydrolysis, likely through a non-processive mechanism of action. Drawing parallels with PET hydrolysis, it may be that enzymatic PA6 deconstruction proceeds via a combination of endo- and exo-lytic bond cleavages[97,98]. As NylC-type enzymes all produced 6-AHA dimer as the major product, if these biocatalysts are selected for further application, it may be that a two-enzyme system is needed to further hydrolyze the dimer to create a

single 6-AHA monomer product. Future enzyme-substrate modeling and structural studies may help further explain the differences in activities of the NylB-type and NylC-type enzymes and shed additional light on their mode of action. It is most likely that NylC$_K$-TS, along with the other enzymes studied, currently deconstructs only a small amount of easily accessible polymer, being unable to progress further once this has been removed, suggesting that a polymer pretreatment step to encourage further depolymerization may be necessary. Our experiments additionally suggest that NylC$_K$-TS activity may be substantially affected by non-productive binding events, indicating that this may be an important area for further investigation when considering future enzyme engineering attempts. However, although the highest percentage depolymerization seen using our most active enzyme, NylC$_K$-TS, was only 0.67 wt% PA6 after 10 days of reaction, the presence of even a small amount of activity in a very stable scaffold provides an important stepping stone from which more active PA6 deconstructing biocatalysts can be realized, as has been demonstrated by the number of successful PETase engineering campaigns following the discovery of numerous PET-active esterases[21,58,72,99,100].

Overall, our analyses indicate that there is significant work ahead to progress enzymatic nylon depolymerization. Of the candidates tested, NylC-type and NylB-type enzymes appeared to be the most specific and active, placing them as the scaffolds with the most potential for enzyme engineering, as has witnessed significant success in the PETase field[21,58,101]. In particular, the NylC$_K$-TS variant developed here may be of particular interest, being both highly thermotolerant and stable over long reaction times. Key characteristics to be targeted for nylonase enhancement, include increasing thermostability and catalytic activity, reduction in non-specific binding, and improving product release specificity. However, given the limited depolymerization extents observed, it is likely that development of substrate pre-processing methods or in situ approaches for improving enzyme access during the reaction will be necessary to access higher extents of depolymerization[102]. It will be important to assess such processes with techno-economic analysis and life cycle assessment to confirm their viability for at-scale beneficial economics and reduced environmental impacts[103,104]. Our study also highlights the need for a more diverse set of starting enzymes for potential nylonase engineering campaigns as, unlike the PETase field, there are few obvious candidates from the currently available literature. Bioprospecting, metagenomic screening, and computational enzyme discovery methods may all offer avenues to different biocatalysts with more suitable PA6 deconstruction characteristics[23,57,105,106]. Such methods, however, should be underpinned by rigorous characterization of enzyme activity on solid PA6, with the LC-MS/MS analysis developed here offering a comprehensive method of assessment for soluble products. In addition, the analytics showcased here could also be expanded to include the deconstruction products of alternative nylon formulations, such as PA6,6, to expand the repertoire of PA-depolymerizing enzymes even further. We are hopeful that the insight offered by our analyses highlight some of the challenges to be overcome and necessary improvements needed to create a viable PA6 biocatalytic recycling strategy.

## Methods
### Materials
PA6 film (0.2 mm thickness, 13.2% crystallinity by DSC, full material characterization available in Supplementary Table 1, Supplementary Fig. 3, product ID: AM30-FM-000200), PA6 powder (particle size: 5–50 μm, 47.6% crystallinity by DSC, full material characterization available in Supplementary Table 3, product ID: AM306010), and PA6 thick film (0.5 mm thickness, full material characterization available in Supplementary Table 3, product ID: AM301400), were sourced from Goodfellow. As a note, we found small differences between batches of PA6 film sourced from Goodfellow (Supplementary Table 4). Crystalline PA6 film (0.2 mm thick, 23.0% crystallinity by DSC, full material

characterization available in Supplementary Table 1, Supplementary Fig. 26) was prepared by cutting up Goodfellow PA6 film into smaller pieces and drying at 150 °C in a vacuum oven overnight. The pieces were hot pressed between Teflon sheets at 250 °C for 1 min at 1,000 psia. The Teflon/nylon film assembly was then placed on a pre-heated hot plate at 200 °C under a pre-heated steel block for 10 min. Finally, the film was dried in a vacuum oven at 200 °C for 2 h. Prior to use, Goodfellow PA6 film and crystalline PA6 film were cut into 0.5 cm×0.5 cm squares, and Goodfellow thick film was cut into 0.32 ×0.32 cm squares. The film squares were then washed in DI water for 3 h at 37 °C, rinsed in DI water once and then dried.

All reagents, chemicals and buffer components were sourced from Sigma-Aldrich, unless specified otherwise. HiC was purchased from Novozymes (Novozym® 51032), subtilisin A, trypsin, papain, and bovine serum albumin (BSA) from Sigma-Aldrich, and α-chymotrypsin from Worthington Biochemical, USA. Butyric anhydride was sourced from TCI chemicals and 1,1,1,3,3,3-hexafluoroisopropanol from Chem IMPEX. The chemical synthesis of N-(4-nitrophenyl)butanamide (N4NB) is detailed in Supplementary Fig. 30 and the corresponding $^1$H NMR spectrum can be found in Supplementary Fig. 31 (analyzed using MNova (Mestrelab Research, version 14.0.0))[107].

### Enzyme identification and homology searches
Literature searches were conducted to find enzymes with observed activity on solid PA6, PA6,6, PA6 oligomers, polyurethane (PUR), or poly(ethylene terephthalate) (PET), considering both scientific reports and patents, with the enzyme names, sources, groups, and reasons for selection detailed in Supplementary Data 1. Amino acid sequences for the selected proteins were taken directly from these reports (Supplementary Data 2). As RoL-WT could not be expressed in E. coli, a previously described mutant which had previous successful E. coli expression (RoL10X) was used[108]. The Ser-His-Asp (SHD) hydrolase group of enzymes was selected from our in-house PET hydrolase library, including two previously reported thermostabilized variants, LCC-ICCG and HotPETase[21,58]. Homology searches for additional NylC-type enzymes and amidases were carried out using UniProt BLAST[66], with NylC$_{p2}$ and NfPolyA as the input protein sequences. The suggested parameters for UniProt BLAST were used (database: UniProtKB, E-threshold: 10, Matrix: Auto-BLOSUM62, Hits: 250), with a homology cut-off of ~50%. From the output sequence homology lists, several candidates were selected that came from potentially thermophilic microorganisms. The phylogenetic tree was inferred by using the Maximum Likelihood method and JTT matrix-based mode, with 1,000 bootstraps[109]. The tree with the highest log likelihood (−4629.08) is shown. Initial tree(s) for the heuristic search were obtained automatically by applying Neighbor-Join and BioNJ algorithms to a matrix of pairwise distances estimated using the JTT model, and then selecting the topology with superior log likelihood value. All positions with less than 90% site coverage were eliminated, i.e., fewer than 10% alignment gaps, missing data, and ambiguous bases were allowed at any position (partial deletion option). There was a total of 88 positions in the final dataset. Evolutionary analyses were conducted in MEGA X[110,111]. The phylogenetic tree was visualized using the Interactive Tree of Life (iTOL) online tool[112].

### Rational mutagenesis of NylCs and NylB
Rational mutations for increasing the thermostability of NylC$_A$ and NylC$_K$ were selected based on positions identified by rational mutagenesis of NylC$_{p2}$ for thermostability by Negoro et al.[49]. The Negoro 4-point mutant, NylC$_{p2}$-TS, (NylC$_{p2}$-D36A/D122G/H130Y,/E263Q) has an increased $T_m$ of 86.5 °C (30 °C increase over the WT, Supplementary Fig. 1). As Negoro et al. found that mutating G111 and L137 of NylC$_{p2}$ to the analogous residues in NylC$_K$ and NylC$_A$ (S111 and A137) significantly decreased thermostability, we hypothesized that the G111 and L137 from NylC$_{p2}$ may be a more stable configuration of these residues, and

hence we introduced them into the scaffolds of NylC$_K$ and NylC$_A$. In addition, the E263Q mutation was included in NylC$_A$ as this change additionally increased the thermostability of NylC$_{p2}$ in the context of D122 and H130, which NylC$_A$ already possesses. This gave the thermostabilized variants NylC$_K$-TS (NylC$_K$-S111G/A137L) and NylC$_A$-TS (NylC$_K$-S111G/A137L/E263Q). The NylB mutation was informed by previous work on the NylB homolog, NylB'[39]. A three point mutant, NylB'-SCY (NylB' R187S/F264C/D370Y), had enhanced 6-AHA dimer hydrolysis ability as compared to the wild-type NylB' enzyme, hence, S187/C264/Y370 were introduced at the analogous positions in NylB to give NylB-SCY. The amino acid sequences for all mutants are presented in Supplementary Data 2.

## Plasmid construction

All protein-encoding genes were codon optimized for expression in *E. coli*, and inserted into either pET-21b, pET-28a or pET-29b expression vectors between NdeI (5′ end) and XhoI (3′ end) restriction sites, dependent on the enzyme, as detailed in Supplementary Data 2. All vectors contain a hexa-histidine tag coding sequence for downstream protein purification. Plasmids were initially synthesized by Twist Biosciences. In-house preparations of the plasmids used in this study have been deposited at AddGene [https://www.addgene.org/Gregg_Beckham/].

## Enzyme expression

All enzymes were expressed in chemically competent BL21 (DE3) *E. coli* cells. Single colonies of freshly transformed cells were grown at 37 °C overnight in 5 mL of LB medium supplemented with the appropriate antibiotic for the expression vector (100 μg/mL ampicillin or 25 μg/mL kanamycin). 2 mL of the resulting culture was used to inoculate 50-100 mL of 2YT medium containing either 100 μg/mL ampicillin or 25 μg/mL kanamycin, as appropriate. Cultures were grown at 37 °C, 190 rpm, to an OD$_{600}$ of 1. Protein production was initiated by the addition of IPTG (final concentration of 0.1 mM), and cultures grown for a further 20 h at 20 °C. Cells were collected by centrifugation at 3,220 × *g* for 10 min and stored at −20 °C until purification.

## Enzyme purification

Cell pellets were resuspended in lysis buffer (pH 7.5, 50 mM Tris-HCl, 10 mM imidazole, 300 mM NaCl, 1.0 mg/mL lysozyme, 10 μg/mL DNase) and subjected to sonication. The resulting lysate was clarified by centrifugation (13,500 × *g* for 15 min) and the soluble fraction applied to Ni-Nta agarose (Anatrace). Unbound proteins were removed with lysis buffer, followed by elution of bound proteins with elution buffer (pH 7.5, 50 mM Tris-HCl, 300 mM imidazole, 300 mM NaCl). Proteins were the applied to 10DG desalting columns (Bio-Rad) and eluted in reaction buffer (100 mM sodium phosphate buffer (NaPi), pH 7.5, 150 mM NaCl). Protein purity was confirmed by SDS-PAGE, with concentrations determined by 280 nm absorbance readings, assuming extinction coefficients detailed in Supplementary Data 2. Protein yields for enzyme purification from a 100 mL *E. coli* culture are detailed in Supplementary Data 1.

## Enzyme thermostability analysis

The thermostabilities of the NylC-TS variants were determined using differential scanning fluorimetry (DSF). For each protein, a 5 μM sample of protein was prepared in reaction buffer and SYPRO Orange dye (provided at a 5000X concentration from the manufacturer), added to a final concentration of 10X. DSF was carried out using a Bio-Rad CFX Connect 96-Real Time PCR system, using the FRET channel for excitation and emission settings, and CFX Manager (version 2.0) for data collection. The temperature was increased with an increment of 0.3 °C/s from 25 °C to 95 °C. The $T_m$ was then determined from the peak of the first derivative of the melt curve from three replicate measurements.

## Screening for activity on N4NB

Activity screens on *N*-(4-nitrophenyl)butanamide (N4NB) were conducted as follows: a reaction mixture of 500 μM N4NB in reaction buffer (100 mM NaPi, pH 7.5, 150 mM NaCl) was arrayed in a 96-well plate. To initiate the reaction, 2 μM of each enzyme was added, the plate sealed with optically clear film, and the release of *p*-nitroaniline (*p*NA) followed by change in absorbance at 410 nm monitored using a Biotek Synergy HT plate reader over the course of 24 h at 30 °C. The concentration of *p*NA was determined by constructing a calibration curve. Enzyme reactions were carried out in triplicate.

## Quantitation of soluble PA6 oligomers by UHPLC-MS/MS operating in dynamic multiple reaction monitoring mode (dMRM)

An analysis method was developed for the quantitation of products from enzymatic deconstruction of PA6 utilizing ultra-high pressure liquid chromatography tandem mass spectrometry (UHPLC-MS/MS). This method employs reverse phase chromatography and dynamic multiple reaction monitoring (dMRM) in positive ion mode using electrospray ionization (ESI) (published on protocols.io, Method 1: [https://doi.org/10.17504/protocols.io.kxygx331dg8j/v1]), Method 2: [https://doi.org/10.17504/protocols.io.6qpvr3k92vmk/v1])[83,111]. Standards were sourced commercially from Sigma Aldrich, Advanced ChemBlocks, and Toronto Research Chemicals. Commercial standards could not be sourced or readily synthesized for 6-AHA tetramer, 6-AHA pentamer, and 6-AHA cyclic-trimer but were still monitored for and quantified from similarly structured compounds, as detailed below. By weight, individual stock standards of all available analytes were prepared at a concentration of 2000 μg/mL using ultrapure water as a diluent. The stock solutions were combined and diluted with ultrapure water to a final concentration of 10 μg/mL. From this mixed standard working solution, a ten-point calibration curve with a quantitation range of 0.01 to 7.0 μg/mL was prepared. A calibration verification standard (CVS) was made at a concentration of 1.0 μg/mL. To check that the linear PA6 oligomers were stable in incubated, buffered solutions, 10 μg/mL of 6-AHA, 6-AHA dimer and 6-AHA trimer chemical standards were incubated individually in reaction buffer (100 mM NaPi buffer, pH 7.5, 150 mM NaCl) at 40-80 °C for 10 days, with samples quenched in MeOH (1 volume sample: 4 volumes methanol), as for enzyme containing reactions. There was no measurable turnover of any of the standards to alternative products, indicating that any changes in linear oligomer concentrations during reactions can be attributed to enzyme action (Supplementary Table 2). Cyclic oligomers can also be considered stable in incubated buffered solutions, as demonstrated in Supplementary Fig. 5B.

Sample and standards were analyzed using an Agilent 1290 series UHPLC system coupled to an Agilent 6470 triple quadrupole mass spectrometer. Ionization was achieved using the Agilent Jet Stream electrospray source operated in positive ionization mode. Optimization of the dMRM transitions for all compounds was executed prior to analysis using authentic standards when available. An enzymatically degraded PA6 sample was used to optimize for the compounds of interest that could not be commercially sourced. Confirmation of the presence of these analytes was performed using high resolution mass spectrometry ESI (HR-MS-ESI). This optimization determined the quantifying and qualifying transitions along with corresponding collision energies and fragmentor voltages (Supplementary Table 5). Data acquisition was conducted using Mass Hunter Data Acquisition software (Agilent, version 10.1).

Initial samples were analyzed with a seven-minute analyte elution protocol (Method 1, analysis method used for data presented in Supplementary Figs. 10, 11, and 13)[113]. Method 1 consisted of sample and standards injected at a volume of 0.5 μL onto a Phenomenex Kinetex 2.1 mm×100 mm, 1.7 μm C18 column held at constant temperature of 40 °C. Chromatography was achieved using mobile phases comprised of (A) 0.1% formic acid in ultrapure water and (B) 0.1% formic acid in

methanol. The gradient program was as follows: $t = 0$ min (A) = 99% and (B) = 1%; $t = 1$ min (A) = 99% and (B) = 1%; $t = 4$ min (A) = 50% and (B) = 50%; at $t = 5$ min (A) = 20% and (B) = 80% returning to initial conditions at $t = 6$ min and holding for 1 minute for a total run time of 7 min. Method 1 was subsequently optimized for a faster run-time of three minutes to give method 2 (analysis method used for all other data presented)[83]. Method 2 employed an injection volume of 0.5 μL onto an Agilent Zorbax Eclipse Plus C18 Rapid Resolution HD column (2.1 ×50 mm, 1.8 μm) held at 40 °C. The same mobile phases were utilized as in method 1 for the following gradient: $t = 0$ min (A) = 99% and (B) = 1%; $t = 0.5$ min (A) = 99% and (B) = 1%; $t = 0.51$ min (A) = 80% and (B) = 20%; at $t = 2$ min (A) = 35% and (B) = 65% returning to initial conditions at $t = 2.01$ minutes and holding for a total run time of 3 minutes. The ESI source parameters for both conditions were identical and optimized as follows; capillary voltage 3 kV, nozzle voltage 0 kV, drying gas temperature 300 °C, drying gas flow 7 L/min, sheath gas temperature 350 °C, sheath gas flow 11 L/min, and lastly the ESI nebulizer pressure was set to 35 psi. Analysis was executed using Mass Hunter Quantitative Analysis for QQQ (Agilent, version 10.1). All calibration curves had an $r^2$ coefficient of ≥ 0.995 with a quadratic curve fit. Quantitation for 6-AHA tetramer and pentamer was performed using the calibration curve from 6-AHA trimer, and 6-AHA cyclic-trimer used the calibration curve from 6-AHA cyclic-dimer. A CVS was injected at least every 20 samples to monitor for instrument drift. An acceptable CVS recovery was determined to be ± 15% of expected concentration. Samples were prepared by a dilution in methanol at a minimum dilution factor of five to facilitate precipitation of buffer salts, to reduce ion suppression and interferences in the source.

### Screening for enzyme activity on PA6 film

The initial activity screens on PA6 films for all enzymes were carried out as followed: reactions contained two squares of PA6 film (13.19 mg ± 0.13 mg PA6, calculated by weighing six pairs of squares and taking the average mass) in reaction buffer (100 mM NaPi buffer, pH 7.5, 150 mM NaCl). 2 μM of enzyme was added to initiate the reaction, giving an enzyme substrate loading of 0.15 mM enzyme/g PA6 film and a substrate loading of 0.65 wt%. Reactions were carried out in triplicate at 40, 50, 60, and 70 °C over the course of 10 days at 180 rpm, with samples taken at 24, 72, 168, and 240 hs. For the comparison of NylC$_K$ and NylC$_K$-TS reactions were caried out with 1 μM enzyme and 13 mg PA6 and incubated at 40–80 °C (0.08 mM enzyme/g PA6, 0.65 wt% substrate loading) with additional time-points taken in the early phase of the reaction. For the characterization of the best performing enzymes, reactions were conducted at 50 °C for NylB'-SCY and 60 °C for Tt-NylC and NylC$_K$-TS to maximize reaction yield while avoiding potentially confounding effects of thermally induced PA6 crystallization observed at higher temperatures. Reactions with variable enzyme or substrate loadings were also carried out in an analogous manner to the initial activity screens with the following changes: for variable enzyme loadings, enzyme concentration was varied from 0.005 μM up to 10 μM per reaction using two squares of PA6 film as the substrate (13 mg), while for changing substrate loadings, 1 μM of enzyme was applied to one to five squares of PA6 film per reaction. For reactions with changing pHs, all buffers were made to 100 mM and supplemented with 150 mM NaCl, using citrate buffer at pH6, NaPi buffer at pH 7-8 and glycine-OH buffer from pH 9-10, and each reaction contained 1 μM of enzyme and two squares of PA6, equating to a final reaction composition of 0.08 mM enzyme/g PA6 film and a substrate loading of 0.65 wt%. Immediately following sampling, all time point samples were quenched in MeOH (1 volume sample: 4 volumes methanol) to denature any active enzyme, filtered using a 0.22 μM PVDF filter plate to remove any solids, and stored at 4 °C prior to analysis. The extent of PA6 film depolymerization by mass was calculated from the release of soluble PA6 depolymerization products as determined by LC-MS/MS.

For the crude cell lysate-based assay, a 50 mL NylC$_K$-TS expressing *E. coli* cell culture was first prepared as above. Following sonication of the resulting cell pellet in lysis buffer, the lysate was clarified by incubation at 60 °C for 30 mins prior to centrifugation. The crude cell lysate was compared to purified NylC$_K$-TS by SDS-PAGE (Supplementary Fig. 6). PA6 deconstruction reactions were then carried out in an analogous manner to those containing purified protein, using 100 μL of the crude cell lysate preparation as the biocatalytic agent.

### Activity assays with nylon oligomers

Activity assays with 6-AHA dimer, 6-AHA trimer and 6-AHA cyclic-dimer were carried out as follows: 50 μM of 6-AHA trimer, 100 μM 6-AHA dimer or 100 μM 6-AHA cyclic-dimer were placed in reaction buffer, pH 7.5 NaPi, 150 mM NaCl. Reactions were initiated by addition of 2 μM enzyme and incubated at the optimal reaction temperature of the described enzyme. Reactions were monitored over the course of 24 h. No enzyme controls showed no detectable hydrolysis of either 6-AHA trimer, 6-AHA dimer or 6-AHA cyclic-dimer over 24 h at reaction conditions.

### Characterization of NylC$_K$-TS activity

For additional characterization of NylC$_K$-TS activity, reactions contained 1 μM enzyme and two squares of PA6 film in reaction buffer and were incubated at 60 °C unless otherwise stated, giving a final reaction composition of 0.08 mM enzyme/g PA6 film and a substrate loading of 0.65 wt%. Crystalline and thick film PA6 reactions were carried out in the same manner using the same mass of the alternative films as the reaction substrates (13 mg, 2 squares of 0.5 cm×0.5 cm and 2 squares of 0.32 cm×0.32 cm, for the crystalline film and thick film, respectively). For reactions with pre-incubated substrate, two PA6 film squares were incubated in reaction buffer for either three or seven days, prior to addition of 1 μM enzyme. For reactions with fresh enzyme added, a NylC$_K$-TS activity assay was carried out for either three or seven days with a time-point sample taken for analysis at the end of reaction, the PA6 squares were removed, sonicated in DI water, rinsed and placed in fresh reaction buffer, 1 μM of fresh enzyme was then added and the reaction allowed to progress with additional product formed added to the value of the time-point sample taken at the end of the first reaction. For reactions with fresh PA6 substrate added, a NylC$_K$-TS activity assay was carried out for either three or seven days with a sample taken for analysis of $T_0$, two new squares of PA6 were then added to the same reaction and reaction progress monitored further. For reactions with bovine serum albumin (BSA), reactions contained 0.01 μM NylC$_K$-TS, two squares of PA6 film (0.8 μM enzyme/g PA6 film and a substrate loading of 0.65 wt%) and 0.5 to 2 mM BSA.

### PA6 substrate characterization

PA6 substrates were prepared as follows: in their unmodified state, as obtained from the manufacturer, or as prepared in-house; washed state, following washing in DI water prior to reactions; or their incubated state where PA6 substrates were incubated in reaction buffer under the same conditions as an enzymatic assay from 40-70 °C (Supplementary Fig. 3). Substrates were also assessed after enzyme reactions, following incubation with 1 μM NylC$_K$-TS at 60 °C for 10 days. Prior to analysis, other than the unmodified substrates, all samples were rinsed in DI water to remove salts and enzymes. All substrates were then dried for 4 h at 50 °C to remove residual water. The thermal stability of nylon samples was assessed by thermogravimetric analysis (TGA) using a Discovery TGA 5500 (TA Instruments). For each run, 6-10 mg of polymer sample was placed in a platinum TGA pan. Samples were heated under nitrogen from ambient temperature to 200 °C at a rate of 4 °C/min, then from 200 °C to 800 °C at a rate of 20 °C/min. TRIOS software (TA Instruments, v5.1.1.46572) was used to characterize the onset temperature of polymer degradation ($T_{D,50}$) and the weight percent of residual char at 800 °C. The mass percent loss and

temperature of the derivative maximum were determined for each mass loss event.

Differential scanning calorimetry (DSC) was used to measure the thermal properties and crystallinity of unmodified and treated nylons. Samples were dried at 40 °C for 24 h in a vacuum oven to remove absorbed water immediately prior to analysis[60]. DSC measurements were performed on a Discovery X3 Differential Scanning Calorimeter (TA Instruments) using 4-8 mg of sample in hermetically sealed aluminum pans (DSC Consumables). Each DSC run consisted of two heating and cooling cycles between 0 °C and 290 °C at a rate of 10 °C/min with 5 min isothermal holds between each heating and cooling ramp. The glass transition temperature ($T_g$), melting temperature ($T_m$), enthalpy of melting ($\Delta H_m$), crystallization temperature ($T_c$), temperature of cold crystallization ($T_c$), and enthalpy of cold crystallization ($\Delta H_c$) for each sample was determined when applicable with TRIOS software (Universal Analysis, v5.4.0.300). Integration bounds and baselines were determined following the procedure described by Khanna and Kuhn[60]. The following equation was used to calculate percent crystallinity, where $\Delta H_m^\circ$, the reference enthalpy of melting, is 230.1 J/g for nylon-6[114].

$$\% \, \text{Crystallinity} = \left( \frac{\Delta H_m - \Delta H_c}{\Delta H_m^\circ} \right) 100\%$$

Gel permeation chromatography (GPC) with multi-angle light scattering (MALS) and differential refractive index (dRI) detectors was used to measure the weight average molar mass ($M_w$), number average molar mass ($M_n$), and molar mass dispersity ($Đ$) values of PA6 samples. Samples were dissolved in 1,1,1,3,3,3-hexafluoroisopropanol (HFIP) with 20 mM of sodium trifluoroacetate (NaTFAc) at a concentration of ~ 5 mg/mL, then filtered through 0.2 μm PTFE syringe filters (Agilent). GPC was conducted using a 1260 Infinity II LC system (Agilent), three PL HFIPgel 250 × 4.6 mm columns, and a matching guard column (Agilent). HFIP supplemented with 20 mM NaTFAc was used as the mobile phase at a flow rate of 0.35 mL/min. Due to HFIP's viscosity, the HPLC column oven was heated to 40 °C to decrease column backpressure. An Optilab T-Rex refractive index detector (Wyatt Technology) and miniDawn TREOS MALS detector (Wyatt Technology) were attached in line. $M_n$, $M_w$, and $Đ$ were calculated with ASTRA (Wyatt Technologies, version 8.2.0), using 0.2375 as the $dn/dc$ of PA6 in HFIP (Wyatt Technologies Database of $dn/dc$ values).

### SEM analysis

SEM analysis was carried out on PA6 films incubated with or without 1 μM NylC$_K$-TS at 60 °C for 10 days. Samples were analyzed by scanning electron microscopy (SEM) on a Hitachi S-4800 High Resolution scanning electron microscope in low- and high-magnification modes at electron voltage = 10 kV, current = ~7 amps, and working distance = 4 mm. Prior to imaging, the samples were coated with a thin layer (10 nm) of Ir using a Cressington plasma sputter-coater (208-HR) under inert (Ar) atmosphere and mounted on an aluminum stage with double-sided C and Cu tapes. Image processing was conducted via publicly available software ImageJ2 (NIH).

### Data visualization

Graphs within the main manuscript and Supplementary Information were created using Microsoft Excel (version 16.80), molecules were drawn using ChemDraw 20.1, and protein structures were visualized using USCF Chimera (version 1.10.1).

### Reporting summary

Further information on research design is available in the Nature Portfolio Reporting Summary linked to this article.

## Data availability

Data supporting the findings of this study are available in the main manuscript and Supplementary Information. Source Data for the figures in the main text are provided in the Source Data file. Data are also available from the corresponding author upon request. Genetic expression constructs for the enzymes expressed in the study have been deposited at AddGene [https://www.addgene.org/Gregg_Beckham/]. Protocols for the described LC-MS/MS analysis methods have been deposited on protocols.io (Method 1: [https://doi.org/10.17504/protocols.io.kxygx331dg8j/v1], Method 2: [https://doi.org/10.17504/protocols.io.6qpvr3k92vmk/v1]). The UniProtKB database used for homology searches is available from the UniProt website [https://www.uniprot.org/blast]. PDB 3AXG, used as the scaffold structure for visualizing NylC$_K$-TS, is available from the Protein Data Bank [https://doi.org/10.2210/pdb3AXG/pdb]. Source data are provided with this paper.

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

## Acknowledgements

ELB, MAI, KJR, CL, RWC, JEG, EE, and GTB were funded by the U.S. Department of Energy, Office of Energy Efficiency and Renewable Energy, Advanced Materials and Manufacturing Technologies Office (AMMTO) and Bioenergy Technologies Office (BETO). This work was performed as part of the BOTTLE™ Consortium and was supported by AMMTO and BETO under contract no. DE-AC36-08GO28308 with the National Renewable Energy Laboratory, operated by Alliance for Sustainable Energy, LLC. ELB and GTB also acknowledge funding from the U.S. Department of Energy, Office of Science, Biological and Environmental Research Office. GR acknowledges funding from the U.S. Department of Energy, Office of Energy Efficiency and Renewable Energy, Bioenergy Technologies Office. We thank Doris Ribitsch and Georg Guebitz for providing an NfPolyA-containing plasmid construct, and John McGeehan for helpful discussions.

## Author contributions

GTB and ELB conceived of the project and designed the study. ELB carried out enzyme identification, expression and purification, conducted all assays, and conducted the data visualization. ELB and EE designed and conducted rational mutagenesis of NylCs. GR contributed synthesis for chemicals and materials. KJR designed the LC-MS/MS analysis method and MAI and KJR ran LC-MS/MS-based analytics. CL characterized the materials used in the study, and RWC conduced SEM imaging. JEG conducted phylogeny analyses. ELB, GR, and GTB, conducted the data analysis. GTB, JLL, and KHK procured funding. The manuscript was written by ELB and edited and approved by all authors.

## Competing interests

ELB, EE, and GTB have filed a patent application on engineered nylonase enzymes. All other authors declare no competing interests.
