## [Peer Review File · Nature Communications]

Reviewers' Comments:

Reviewer #1:

Remarks to the Author:

The 'superiority' of the PETase from *Ideonella sakaiensis* for enzymatic polyethylene terephthalate degradation is a baffling misunderstanding that has been surprisingly persistent in the literature, despite the great progress made using other hydrolases, esp. the engineered cutinase LCC-ICCG. This is more surprising given that LCC was described in 2012, four years before the PETase from *I. sakaiensis* (and on top of that, cutinases were already known to hydrolyze PET back in 2005). This serves to demonstrate how badly focus and resources may be allocated once interest in an interesting-but-inferior enzyme gets out of hand, with no solid comparisons to guide the scientific community. Even more surprisingly, this is an ongoing situation, as explained in a recent paper by the groups of Bornscheuer, Weber, and Marty (Arnal et al., <https://doi.org/10.1021/acscatal.3c02922>). This passage from their abstract makes the point: "A large set of promising hydrolases that depolymerize PET have been found and enhanced by worldwide initiatives using various methods of protein engineering. Despite the achievements made in these works, it remains difficult to compare enzymes' performance and their applicability to large-scale reactions due to a lack of homogeneity between the experimental protocols used." I think the authors should cite this paper and that by Wei et al. (doi.org/10.1021/acscatal.1c05856), which makes a similar point.

That direct comparison of previously described nylonases is challenging is therefore perhaps the most important point made by Bell et al. (Line 54). A great feature of their manuscript is that this issue is addressed and that previously described enzymes are directly compared, using analytics that are broadly available and applicable to future investigations of novel nylonases. Perhaps the greatest strength of the manuscript is thus that it sets the stage for proper comparable research early on; it would be most disappointing if research on nylon degradation gets as derailed by an overpublicized enzyme as research on PET degradation was. In this context, the most likely criticism of the paper, that it mostly reports an LC-MS/MS assay used to study mostly known nylonases, appears invalid to me. That said, the contribution of engineering the most active nylonase so far described (NylCK-TS) is not emphasized clearly enough. A clearer comparison to the wild-type enzyme would further strengthen the paper and make it more suitable for publication in *Nature Communications*. Below are several minor comments that the authors may consider addressing if it is accepted for publication or if they are given the opportunity to revise their manuscript.

Line 6; "PA6" was not defined in the abstract

Line 14; For a thermostabilized protein, it would be good to also give the increase in T_m or the wild-type T_m

Line 15; If possible, clarify that "substrate-based limitations" means that you think the bonds are inaccessible to enzymatic hydrolysis

Line 38; Also citing reference 97 (Branson et al.) here would make sense. Please give another example of what "similar approaches with other polymers" could mean (or is it limited to PET, nylon, and PU?)

Line 45; "serine-reactive hydrolase" sounds like an unusual term and using "reactive-serine hydrolase" or simply "serine hydrolase" would be clearer

Line 47; please explain the term "promiscuous cutinases", "cutinases" would likely be clearer. The EC number given is for carboxylic ester hydrolases, it is EC 3.1.1.74 for cutinases

Line 48; "proteases (EC 3.4.21)" might be too limited. Later papain is mentioned (Figure 1) and this is a cysteine protease (EC 3.4.22)

Line 52; For "Previous studies have necessarily used a wide variety of analyses and substrates" it would be nice to have a short summary or at least a few examples to demonstrate this diversity

Line 61; "over 30" could be replaced with the actual number of enzymes studied

Line 63; "high thermostability, a desirable characteristic": please discuss which temperature is most likely to be ideal for enzymatic degradation

Line 64; "high-throughput" is vague and context-dependent. Please estimate the throughput (number of samples per hour or day) that can be processed (even if the reader could calculate that based on the approximately 3 min runtime). Please also discuss limitations in terms of sample properties, e.g., how would performance be affected if crude lysates rather than purified proteins were used for degradation experiments? This has one of the greatest impacts on throughput

Line 66; Please explicitly state, for clarity, that these compounds can all be present and quantified simultaneously (in the same sample)

Line 71; "Ntn hydrolases appear to be the most active" could be misleading since, while true for the small set of about 30 enzymes tested, the most active enzymes to be discovered in future studies may not be Ntn hydrolases

Line 72; "nylonases only hydrolyze a small portion of available PA6": please interpret in the context of published work (i.e., was higher or lower degradation reported before?). Please also consider referring to the characterization of material properties later in the manuscript/SI, and mention here that addition of fresh PA does result in more product formation, demonstrating that this is not an effect of enzyme failure. While these are discussed later in the paper, not mentioning it here could cause confusion

Figure 1; Why are only amino groups ionized and the acids not? For the NylC-type proteins, the "Nter" might be misleading, since the N-terminal threonine is drawn, the other end is rather the C-terminus ("Cter")

Line 93; "PURases" is not a common term, "polyurethanases" is much more common and clearer

Line 103; For "homology searches", please mention what the minimum identity for enzymes to be considered was

Line 104; Check the author format for ref 56 (not "The UniProt Consortium et al")

Line 109; Please edit "serine proteases" since papain is not a serine protease

Line 119; Comma before "was" is confusing

Line 126; "N-butyl-4-nitroaniline" is not the correct name for the substrate, this error appears a few times in the manuscript and SI (including the heading for Figure 2)

Line 128; "Both the amidases" > "The amidases"

Line 130; For "amide hydrolysis activity", it is important to state either how much enzyme was used for the assays or to express the values as specific activities. While this information is found later in the manuscript, it is still critical here

Line 134; "highest performers" may be misleading, since these are just the highest in the NylC-type and SHD-hydrolase groups

Line 139; While this strategy is generally valid, the authors point out later that there isn't a strong correlation between NB4N hydrolysis and nylonase activity. Therefore, please comment on the possibility that promising enzymes may have been excluded since the chromogenic substrate is as different from nylon as it is from the natural substrates for proteases

Line 154; Please comment on the throughput of the assay rather than just calling it "high-throughput" (in the following paragraph, not necessary in the heading)

Line 165; While current enzymes are too slow for this to be relevant, the speed of the assay may be very valuable for kinetics/following changes in distributions of products when faster enzymes are discovered

Line 167; Did the calibration standard contain all the analytes shown in Figure 3?

Figure 3; Please state which enzyme was used in this reaction for the representative chromatogram

Line 188; Should "differential scanning crystallinity" be "differential scanning calorimetry"?

Line 194; Please explain "6.5 μ M enzyme/mg washed PA6 film". This format for the amount of enzyme is used several times throughout the paper but it is unclear. For example, in line 221, "the reactions contained 2 μ M enzyme (6.5 μ M enzyme/mg PA6), 13 mg PA6" is hard to understand. If there is 13 mg PA6 and 6.5 μ M enzyme/mg, one would expect 84.5 μ M, not 2 μ M. It also does not make sense to increase enzyme concentration as a function of the mass of PA6, it would make more sense that the amount of enzyme was increased. Please clarify this and adjust the values/units throughout the manuscript if necessary

Line 195; "two squares of 0.5 x 0.5 cm PA6 film=13 mg" Please state the density of the material used (assuming 1.14 g/ml, the calculated mass would be 11.4 mg, not 13 mg)

Figure 4; "NyCp2" should be "NylCp2"

Line 215; list the enzymes/colors in the same order as in the figure

Line 220; "a grey circle" is used throughout the manuscript where "grey circles" would sound more appropriate

Line 227; "70°C"; inconsistency regarding the space before the °C symbol, please check throughout the manuscript

Line 239; "Despite the promise..." Here it is not clear whether the 0.67 wt% in "monomer release at 70 °C after 10 days only equates to 0.67 wt% PA6" refers to 6-AHA or 6-AHA equivalents, since NylCK-TS does not degrade the dimer (Figure S12C). The amount for 6-AHA equivalents would be of interest here

Line 260; "which produces only 49.8 μM of 6-AHA" Here, again, it is unclear if "6-AHA" refers to the monomer or to "6-AHA equivalents" (looks like equivalents in Figure 4B)

Line 271; "of over 10 days" > "over 10 days"

Line 277; "Supplementary Fig. 15" It is not clear from part A of the figure whether there is no data for day 10 for GatA+ or whether the dimer was undetectable. Same for GatA+ for part B of the figure, but for all days. For part A, please explain why dimer concentrations are increased on days 3 and 7, relative to day 1, while on day 1 it is much lower than in the no-enzyme control

Figure 5; Part A: The legends on the left and right of the figure do not obviously represent the entire figure, it would be clearer if they were both moved to the right of the figure

Line 318; "data in panel A and B" > "data in panels A and B"

Line 354; My most serious question is whether the accessible fraction is dependent on the surface area of the plastic used; it likely is. This leads to the question of why the authors did not investigate the effects of grinding or other means of increasing surface area (e.g., making nanoparticles) on the rate and extent of degradation of the polymer. While it is not necessary to add such experiments to the current paper, the issue must be discussed

Line 411; doi.org/10.1038/s41467-018-07488-0 may be relevant to readers

Line 416; "0.18 mg enzyme/g PA6" is not in the same format as other amount enzyme/amount PA values in the manuscript, it would be easier to compare if it were

Lines 427-431; Comparison to the wild-type would be very helpful here. That this enzyme (NylCK-TS), engineered in the current work, is the most active of the nylonases tested, increases the relevance of the paper to a journal like Nature Communications. However, the increase in performance (thermostability and 6-AHA production at various temperatures) must be clearly stated relative to the wild-type

Line 741; BL21(DE3)

Line 761; "concentration of 10X" is unclear

Line 766; Please check substrate name

Lines 819-820; Please check enzyme amount, as commented on earlier in this report

Line 832; "quenched in 5X MeOH" does this mean five volumes of methanol?

Supplementary Fig. 1: Comparisons to the wild-type proteins would be valuable

Supplementary Fig. 2: Fix substrate name

Supplementary Fig. 5: "grey circles"

Supplementary Fig. 12: "grey circles"

Supplementary Fig. 13: There are no grey circles in this figure

Supplementary Fig. 14: An AlphaFold model is not a "crystal structure"

Supplementary Fig. 15: See comment to line 277

Supplementary Fig. 16: Data points are represented by squares, triangles, and circles, not just circles

Supplementary Fig. 18: Investigation of the effect of increasing surface area would have been very interesting (grinding or making nanoparticles)

Reviewer #2:

Remarks to the Author:

In this article, the authors investigated the enzymatic hydrolysis of Nylon-6 (PA6). The reviewers posit that the authors have furnished readers with novel insights on the following aspects:

- (1) The authors achieved amino acid substitutions toward NyICK, distinct from the NyICp2 base GYAQ mutant extensively scrutinized by Negoro's group, with the aim of enhancing the heat resistance property.
- (2) The analysis encompassed enzymatic degradation of commercially available PA6 film.

Regrettably no novel superfamily was unveiled, though notwithstanding the authors' efforts to screen a new enzyme superfamily capable of hydrolyzing PA6 from the vast expanse of nature. While acknowledging the pioneering endeavors of Negoro's group in the realm of nylon hydrolytic enzymes, the authors ultimately discerned only the same NyIB- and NyIC-type enzymes. Furthermore, the degradation test for the PA6 films were on par with those previously reported by Negoro's group.

Hence, while the reviewer does not impugn the validity of the authors' work, the reviewer harbors reservations about whether the authors' findings constitute a sufficiently remarkable advancement to warrant publication in Nature Communications. The reviewer strongly recommends that, after the addition of biochemical characterization test for the newly discovered NyICK mutants, it may be judicious to resubmit the papers such as FEBS J or Sci. Rep. and so on.

Additionally, to enhance the paper's caliber, the reviewer implores the authors to reevaluate the following:

(i) Negoro's group introduced a method for quantifying reaction rates employing thin-layered films of nylon 6 or 66 on 96-well plates. Astonishingly, this method was not cited in the paper, constituting a grave omission.

Reference: Nagai, Keisuke, et al. (2014), 'Enzymatic hydrolysis of nylons: quantification of the reaction rate of nylon hydrolase for thin-layered nylons', *Appl. Microbiol. Biotechnol.*, 98 (20), 8751-61.

(ii) During the screening of nylon hydrolytic enzymes, the authors employed N-butyl-4-nitroaniline as the substrate, a compound markedly distinct from nylon 6 monomer unit structure. Consequently, the reviewer harbors doubts about the efficacy of this substrate for enzyme screening, as it may be more suitable for the screening of NylB-type enzymes (exo-type), but ill-suited for the screening of endo-type enzymes like NylC. The endo-type NylC enzymes do not recognize this substrate, and it remains uncertain whether effective screening has been achieved. Demonstrating the validity of this screening system is imperative. From this perspective, the reviewer strongly suggests that adopting the aforementioned thin-layered film screening system could potentially unearth previously concealed enzymes.

(iii) The authors are urged to validate the amino acid numbering of NylC mutants. For instance, S112G should be corrected to S111G, and E262Q should similarly be adjusted to E263Q.

Reviewer #3:

Remarks to the Author:

This manuscript by Bell and coworkers reports a comprehensive and systematic investigation of the enzymatic degradation of nylon. The work is well motivated and presents a high level of novelty. Thus, based on the chemical structure of Nylon, bioprocessing of Nylon waste indeed represents a promising case – particularly in the light of recent progress within enzyme-based monomer recovery from polyester waste. Yet, the literature on nylon hydrolases is sparse and quite incompatible with respect to methods and strategies.

To the best of my knowledge this work is the first systematic study where a larger group of potential nylon hydrolases have been identified, organized in different groups, produced, engineered, and characterized in a systematic way for direct comparison. This represents a huge work and an important step forward in our understanding of enzymatic nylon degradation.

The manuscript is clearly written with a comprehensive empirical material presented in compressed but well-composed figures as well as a substantial supplementary material.

I recommend the publication of this work.

A few comments for consideration.

Line 188: I guess the authors mean differential scanning calorimetry.

Line 235-240. Are the oligomers discussed here stable in enzyme-free solutions or could the distribution of the different oligomers be affected by non-catalyzed hydrolysis?

Line 259: I did not understand what was meant by "concerted depolymerization".

Line 333: I found it remarkable that BSA (at a 10:1 or so ratio to the enzyme) increases the specific activity dramatically (some 100-fold). The interpretation (lines 334-5) is: "that while NylCK-TS may be affected by non-specific binding events, this is not a main factor". That seems like an understatement to me. If the effect of BSA is related to nonspecific adsorption to the reaction vial or non-productive binding to the PA surface, one could argue that the remaining enzyme population was two order of magnitude more active than what is captured by average numbers. I might be that I have misunderstood this, but if not, this would be an observation to bring on in future attempts to find or

engineer better nylon hydrolases. I suggest the authors elaborate on this and its relationship e.g. to the dose response curves in Fig. 5B.

Reviewer #4:

Remarks to the Author:

I was tasked with examining the applied mass spectrometric protocols employed to quantify the soluble fractions resulting from the enzymatic PA6 degradation. The authors have developed a careful protocol based on direct calibration of available standards. The used methodology is in line with best practice in the field and I could not identify any critical flaws or weaknesses in their approach.

Reviewer comments are provided in black font. Our responses are shown in blue font. Changes to the manuscript are shown in red font.

Reviewer #1 (Remarks to the Author):

The ‘superiority’ of the PETase from *Ideonella sakaiensis* for enzymatic polyethylene terephthalate degradation is a baffling misunderstanding that has been surprisingly persistent in the literature, despite the great progress made using other hydrolases, esp. the engineered cutinase LCC-ICCG. This is more surprising given that LCC was described in 2012, four years before the PETase from *I. sakaiensis* (and on top of that, cutinases were already known to hydrolyze PET back in 2005). This serves to demonstrate how badly focus and resources may be allocated once interest in an interesting-but-inferior enzyme gets out of hand, with no solid comparisons to guide the scientific community. Even more surprisingly, this is an ongoing situation, as explained in a recent paper by the groups of Bornscheuer, Weber, and Marty (Arnal et al., <https://doi.org/10.1021/acscatal.3c02922>). This passage from their abstract makes the point: “A large set of promising hydrolases that depolymerize PET have been found and enhanced by worldwide initiatives using various methods of protein engineering. Despite the achievements made in these works, it remains difficult to compare enzymes’ performance and their applicability to large-scale reactions due to a lack of homogeneity between the experimental protocols used.” I think the authors should cite this paper and that by Wei et al. (doi.org/10.1021/acscatal.1c05856), which makes a similar point.

We thank the reviewer for their comments into how lessons learned in the PETase field can help inform and promote high-quality research into other plastic-deconstructing enzymes, such as nylonases. We fully agree that robust comparisons between nylonases prior to extensive research or enzyme engineering being conducted on individual enzymes is a positive step forwards for the field.

In response, we modified the third paragraph of the Introduction to more strongly advocate for the need for robust comparisons between enzymes, and we added the suggested references from the reviewer.

That direct comparison of previously described nylonases is challenging is therefore perhaps the most important point made by Bell et al. (Line 54). A great feature of their manuscript is that this issue is addressed and that previously described enzymes are directly compared, using analytics that are broadly available and applicable to future investigations of novel nylonases. Perhaps the greatest strength of the manuscript is thus that it sets the stage for proper comparable research early on; it would be most disappointing if research on nylon degradation gets as derailed by an overpublicized enzyme as research on PET degradation was. In this context, the most likely criticism of the paper, that it mostly reports an LC-MS/MS assay used to study mostly known nylonases, appears invalid to me. That said, the contribution of engineering the most active nylonase so far described (NylCK-TS) is not emphasized clearly enough. A clearer comparison to the wild-type enzyme would further strengthen the paper and make it more suitable for publication in *Nature Communications*. Below are several minor comments that the authors may consider addressing if it is accepted for publication or if they are given the opportunity to revise their manuscript.

We thank the reviewer for their summary of the motivation for – and potential value of this work – to the research community. We agree that studies of this type are necessary to help provide information to guide further research efforts.

We also agree with the reviewer that the comparison of NylCK and NylCK-TS should be more prominent in the current manuscript, as the ability to engineer an enzyme is an important factor to consider in the selection suitable nylonases for further study.

We therefore added an additional figure in the main text directly comparing NylCK and NylCK-TS (**Fig. 5**), with an associated paragraph describing the findings. A new full dataset of comparisons of the two enzymes from 40–80 °C is now presented in **Supplementary Fig. 16**, alongside a structural model highlighting where mutations have been made (**Fig. 5A**). We included additional experiments with the enzymes at 80 °C to show case our engineered enzyme’s thermostability.

Line 6; “PA6” was not defined in the abstract

Overall, we thank the reviewer for their detailed reading of the manuscript. All suggested minor changes in the following pages have been addressed in the manuscript and are detailed in red text below.

This has been defined as “enzymatic nylon-6 (PA6) hydrolysis”.

Line 14; For a thermostabilized protein, it would be good to also give the increase in T_m or the wild-type T_m

The T_m increase compared to WT has been added: “(an N-terminal nucleophile (Ntn) hydrolase, $T_m=87.4$ °C, 16.4 °C higher than the wild-type).”

Line 15; If possible, clarify that “substrate-based limitations” means that you think the bonds are inaccessible to enzymatic hydrolysis

This has been reworded to: “Reactions fail to restart after fresh enzyme addition, indicating that substrate-based limitations such as restricted enzyme access to hydrolyzable bonds, prohibit more extensive deconstruction.”

Line 38; Also citing reference 97 (Branson et al.) here would make sense. Please give another example of what “similar approaches with other polymers” could mean (or is it limited to PET, nylon, and PU?)

The original phrasing was confusing – we were aiming to convey how we would like to use similar approaches for other polymers, not that similar industrial scale approaches, as have been demonstrated for PET, are already available.

The text has been updated to: “Such successes in biocatalytic PET depolymerization have motivated interest in emulating similar approaches with other polymers, with interests mainly focused on other C–O and C–N linked plastics,^{23,24} including nylons.”

This statement now includes the Branson *et al.* reference and a recent review from Tournier *et al.* (doi.org/10.1021/acs.chemrev.2c00644), which more comprehensively explores the types of polymers the community can likely recycle enzymatically.

Line 45; “serine-reactive hydrolase” sounds like an unusual term and using “reactive-serine hydrolase” or simply “serine hydrolase” would be clearer

This has been revised to “serine hydrolase”.

Line 47; please explain the term “promiscuous cutinases”, “cutinases” would likely be clearer. The EC number given is for carboxylic ester hydrolases, it is EC 3.1.1.74 for cutinases

This has been revised to “cutinase”.

Line 48; “proteases (EC 3.4.21)” might be too limited. Later papain is mentioned (Figure 1) and this is a cysteine protease (EC 3.4.22).

The text has been updated to remove ambiguity to: “Numerous other hydrolases have potential PA depolymerization activity, including Ser-(*cis*)Ser-Lys containing amidases (EC 3.5.1.4), cutinases (EC 3.1.1.74) using the canonical Ser-His-Asp (SHD) catalytic configuration and multiple proteases using either serine or cysteine residues as catalytic nucleophiles (EC 3.4.21 & 3.4.22).^{27,40–48}”

Line 52; For “Previous studies have necessarily used a wide variety of analyses and substrates” it would be nice to have a short summary or at least a few examples to demonstrate this diversity

We have added two sentences to address this in the Introduction: “For instance, while some studies use proxy measurements to evaluate nylon-active enzymes, such as looking for polymer surface changes via rising-height analysis,^{43,46} or using indicator compounds to identify the release of amine-containing molecules,^{49–51} others use soluble substrate analogues to probe enzyme kinetics,⁵² or look for changes in thickness of self-cast nylon films following enzyme application.⁵³ However, the large diversity of experimental set-ups and substrates used across such studies, makes a direct comparison of previously described enzymes challenging.”

Line 61; “over 30” could be replaced with the actual number of enzymes studied

This has been revised as suggested.

Line 63; “high thermostability, a desirable characteristic”: please discuss which temperature is most likely to be ideal for enzymatic degradation

This has been clarified to: “Within the enzyme set, we also included engineered enzymes with high thermostability, a desirable characteristic to enable polymer deconstruction near the glass transition temperature (T_g) of the plastic (T_g PA6 = ~ 50-55 °C),⁶⁰ whereby substrate accessibility may be increased.⁶¹”

Line 64; “high-throughput” is vague and context-dependent. Please estimate the throughput (number of samples per hour or day) that can be processed (even if the reader could calculate that based on the approximately 3 min runtime). Please also discuss limitations in terms of sample properties, e.g., how would performance be affected if crude lysates rather than purified proteins were used for degradation experiments? This has one of the greatest impacts on throughput

We thank the reviewer for raising these points. We agree that the ability to apply our analysis methods to crude lysates is an interesting question and would increase the applicability of this technique. We therefore ran additional experiments comparing purified NylC_K-TS to assays with NylC_K-TS-containing cell lysate prepared via cell pellet sonication without subsequent purification. Fortunately, our analysis method is unaffected by using cell lysate relative to purified protein, with reactions of purified protein and cell lysate being comparable:

The findings of the lysate experiments are detailed in **Supplementary Figs. 6-7** and the following sentence has been added to the main text: “Although the nylonase deconstruction assays presented below were all conducted with purified proteins, the LC-MS/MS analytical method is unaffected by instead using crude cell lysate as the biocatalytic agent (**Supplementary Fig. 6**), with a similar product release profile observed as compared to using purified protein (**Supplementary Fig. 7**).”

In response to this comment, we also added additional information to quantify the throughput of the method and number of samples that can be processed continuously:

We added the following sentence to the “Development of a high-throughput solid PA6 depolymerization analysis method by LC-MS/MS” section: “The use of a multisampler allows eight 96-well plates to be analyzed without manual intervention, with around 20 samples being analyzed per hour.”

Line 66; Please explicitly state, for clarity, that these compounds can all be present and quantified simultaneously (in the same sample)

We have clarified this sentence to “... using a high-throughput nylon depolymerization analysis method based on liquid chromatography with tandem mass spectrometry (LC-MS/MS), which accurately and simultaneously detects and quantifies linear-oligomers of PA6 up to pentamers, and cyclic-oligomers up to trimers in a single sample.”

Line 71; “Ntn hydrolases appear to be the most active” could be misleading since, while true for the small set of about 30 enzymes tested, the most active enzymes to be discovered in future studies may not be Ntn hydrolases

We have clarified this sentence to “Ntn hydrolases appear to be the most active of the enzymes examined here.”

Line 72; “nylonases only hydrolyze a small portion of available PA6”: please interpret in the context of published work (i.e., was higher or lower degradation reported before?). Please also consider referring to the characterization of material properties later in the manuscript/SI, and mention here that addition of fresh PA does result in more product formation, demonstrating that this is not an effect of enzyme failure. While these are discussed later in the paper, not mentioning it here could cause confusion

We thank the reviewer for their suggestions. It is difficult from the available literature to quantify as a percentage how much nylon has been deconstructed in previous studies due to the wide variety of analysis techniques and substrates used. The most comparable data suggest that NylC_{p2} depolymerizes 1% of a nylon-6 powder suspension (<https://doi.org/10.1007/s00253-014-5885-2>), but this percentage depolymerization was estimated from the concentration of released amine groups, so it is unclear how accurate this figure is considering that we show multiple different oligomers of PA6 of different sizes are released, and also how fair a comparison this is to our work focusing on a film, considering the differences in surface areas and polymer characteristics.

We have therefore instead added the range of depolymerization extents seen as a percentage, for the best enzymes we found from each group, to more clearly define what we mean by small portion: “We present evidence that

even the best nylonases tested only hydrolyze a small portion of available PA6, with between 0.1-0.7% of the PA6 depolymerized by the best enzymes we examined.”

We have added a sentence to the last paragraph of the introduction to highlight the importance of the materials characterization work carried out within the study: “Product release analysis was coupled with rigorous characterization of the PA6 materials used in the study, both pre- and post- enzymatic deconstruction.”

We have added a sentence to the last paragraph of the introduction to more clearly explain what our enzyme/substrate addition experiments show: “For NylC_K-TS, no further depolymerization was observed after fresh enzyme addition; however additional product release was observed following addition of fresh PA6, indicating that enzyme stability does not explain low depolymerization extents, and instead low extents of deconstruction may be considered a substrate-based phenomenon.”

Figure 1; Why are only amino groups ionized and the acids not? For the NylC-type proteins, the “Nter” might be misleading, since the N-terminal threonine is drawn, the other end is rather the C-terminus (“Cter”)

We thank the reviewer for pointing out the omission of the negative charge on the Asp residue.

We edited the image to include the charge and have changed the “Nter” to a wavy line to indicate connection to the rest of the protein structure.

Line 93; “PURases” is not a common term, “polyurethanases” is much more common and clearer

This has been modified throughout to polyurethanase/s.

Line 103; For “homology searches”, please mention what the minimum identity for enzymes to be considered was

We have updated the Methods section to include the parameters used for the search.

Line 104; Check the author format for ref 56 (not “The UniProt Consortium et al”)

This has been modified to “The UniProt Consortium” as recommended on the UniProt website.

Line 109; Please edit “serine proteases” since papain is not a serine protease

This has been modified to “serine and cystine proteases” throughout the manuscript.

Line 119; Comma before “was” is confusing

The comma has been removed.

Line 126; “N-butyl-4-nitroaniline” is not the correct name for the substrate, this error appears a few times in the manuscript and SI (including the heading for Figure 2)

We thank the reviewer for pointing this out.

The compound is now referred to as *N*-(4-nitrophenyl)butanamide or N4NB throughout.

Line 128; “Both the amidases” > “The amidases”

This has been updated to “The amidases”.

Line 130; For “amide hydrolysis activity”, it is important to state either how much enzyme was used for the assays or to express the values as specific activities. While this information is found later in the manuscript, it is still critical here

We added the substrate and enzyme concentrations, and reaction conditions, to the indicated sentence: “To test the ability of each enzyme for amide bond hydrolysis, candidates were incubated with *N*-(4-nitrophenyl)butanamide (N4NB), a small molecule surrogate that contains a butyramide motif akin to the amide bonds in PA6 (0.5 mM N4NB, 2 μM enzyme, 30 °C in reaction buffer (100 mM sodium phosphate buffer, pH 7.5, 150 mM NaCl), Fig. 2, Supplementary Fig. 2A-I).”

Line 134; “highest performers” may be misleading, since these are just the highest in the NylC-type and SHD-hydrolase groups

We have clarified this to: “The highest performers in these two groups...”.

Line 139; While this strategy is generally valid, the authors point out later that there isn't a strong correlation between NB4N hydrolysis and nylonase activity. Therefore, please comment on the possibility that promising enzymes may have been excluded since the chromogenic substrate is as different from nylon as it is from the natural substrates for proteases

We thank the reviewer for raising this valid point. The only enzymes excluded at this stage were the proteases. Hence, to rule out that using this system high performing enzymes may be erroneously removed from further analysis due to substrate preference differences, we performed additional experiments to test the proteases in PA6 film hydrolysis experiments. This revealed that the proteases we tested had no detectable nylon hydrolysis ability as determined by measuring the soluble nylon oligomer release. It therefore appears although good activity with N4BN does not necessarily indicate good activity on PA6, poor activity on N4BN can be considered an indicator of poor activity on PA6. We do agree however that the lack of soluble product release we see in our solid PA6 assays may not indicate a complete lack of polymer hydrolysis activity by the proteases, but rather it may be that these enzymes produce much longer oligomers that are not detected and quantified by our assay or they may cleave polymers mid-chain, so no soluble products are released. We have therefore included a note to that effect to highlight potential limitations of our method:

We altered the text to more clearly highlight where N4BN hydrolysis experiments can add value to nylonase discovery campaigns and stress that the best indicator of nylonase activity is thorough testing on a real PA6 substrate in the Conclusion: "Perhaps unsurprisingly, there was little correlation between the capacity for an enzyme to hydrolyze the soluble N4NB substrate relative to solid PA6 deconstruction: our findings indicate that soluble substrates such as N4NB may be used to indicate poor nylonase activity, but are not good indicators of high nylonase performance on solid PA6."

The data for the protease experiments are provided in **Supplementary Fig. 12**, with the associated text as follows: "No measurable PA6 depolymerization activity was seen for any of the proteases or miscellaneous category enzymes tested here (**Supplementary Fig. 12 & 13**). It is possible that these enzymes may cleave polyamide, but release products of a higher molecular mass than what our analysis method detects; however, a lack of subsequent depolymerization down to smaller molecular mass oligomers or monomers suggests that these six enzymes may not be the best choice for further nylonase development."

Line 154; Please comment on the throughput of the assay rather than just calling it "high-throughput" (in the following paragraph, not necessary in the heading)

We thank the reviewer for their suggestion. We believe we are correct in referring to the method as high-throughput as it can be carried out in a 96-well plate format, without additional human intervention for sample application, and we can collect quantification of eight analytes from a single sample in three minutes.

As described above, we added information to the "Development of a high-throughput solid PA6 depolymerization analysis method by LC-MS/MS" section to clarify the throughput of the technique: "The use of a multisampler allows eight 96-well plates to be analyzed without manual intervention, with around 20 samples being analyzed per hour."

Line 165; While current enzymes are too slow for this to be relevant, the speed of the assay may be very valuable for kinetics/following changes in distributions of products when faster enzymes are discovered

We thank the reviewer for their comment; however, we believe there may be a misunderstanding of the state of the samples prior to analysis and the analytical method. The analytical method is *ex-situ*, meaning that the time for analyzing the samples does not affect the distribution of products seen. Following sample collection from the enzyme reaction at the defined time point, the reactions are quenched in 5X MeOH to stop the reaction and denature the enzyme and filtered to remove any potential solid substrate. Therefore, at the point of sample collection, the reaction is paused and hence product quantification will not be affected by analytical run time. In addition, all the reaction products quantified are stable in 5X MeOH for at least 2 weeks as tested using product standards prepared in an analogous manner. If we have misunderstood the meaning of this comment, we would be happy to follow this up in any additional revisions.

We added additional information to the Methods section to make this clearer: "Immediately following sampling, all time point samples were quenched in MeOH (1 volume sample: 4 volumes methanol) to denature any active

enzyme, filtered using a 0.22 μM PVDF filter plate to remove any solids.”, and have added the fact that the method is *ex-situ* to the “Development of high-throughput solid depolymerization analysis” section.

Line 167; Did the calibration standard contain all the analytes shown in Figure 3?

The calibration verification standard (CVS) contains a mixture of all analytes which were available as commercial standards, with each compound present at a concentration of $1.0 \mu\text{g mL}^{-1}$, as detailed in the Methods section and the protocol.io file method walkthrough (attached to our submission as a pdf file).

We have referenced this method again when we describe the calibration standard to aid clarity.

Figure 3; Please state which enzyme was used in this reaction for the representative chromatogram

We updated the sentence to: “Representative LC-MS/MS chromatogram following enzymatic hydrolysis of PA6 (0.01 μM NylC_K-TS, 13 mg PA6, 60 °C), showing the eight compounds that were detected and quantified.”

Line 188; Should “differential scanning crystallinity” be “differential scanning calorimetry”?

This was a mistake – we have updated this to differential scanning calorimetry.

Line 194; Please explain “6.5 μM enzyme/mg washed PA6 film”. This format for the amount of enzyme is used several times throughout the paper but it is unclear. For example, in line 221, “the reactions contained 2 μM enzyme (6.5 μM enzyme/mg PA6), 13 mg PA6” is hard to understand. If there is 13 mg PA6 and 6.5 μM enzyme/mg, one would expect 84.5 μM , not 2 μM . It also does not make sense to increase enzyme concentration as a function of the mass of PA6, it would make more sense that the amount of enzyme was increased. Please clarify this and adjust the values/units throughout the manuscript if necessary

We thank the reviewer for pointing out this miscalculation.

The enzyme loadings have been corrected throughout the manuscript and the units standardized to mM enzyme/g PA6 (to better aid comparison to mg/g measures).

We still believe the mg enzyme/g PA6 may be a useful conversion for readers to compare to other studies and substrates where this can be a common measure.

To avoid confusion, when we mention the mg enzyme/g substrate, it is presented in addition to the mM/g measure, rather than as a replacement, and when we mention enzyme loadings the order in which they are mentioned in is now consistent throughout the manuscript.

We are unclear as to what is meant by the comment on increasing enzyme concentration as a function of mass. Two sets of experiments are presented in the manuscript where either the concentration of enzyme is increased with the mass of substrate remaining constant, or where the mass of substrate is increased with the concentration of enzyme remaining constant. There were no experiments carried out where these variables were changed in conjunction.

We rephrased the line in the “Assessment of potential nylonases” section to define the substrate loading and enzyme/substrate loading terms to aid clarity, in case this was the point of confusion: “Enzyme reactions were carried out in 100 mM sodium phosphate buffer (NaPi), pH 7.5, 150 mM NaCl, and contained 2 μM enzyme and 13 mg PA6 (two squares of 0.5 x 0.5 cm PA6 film), representing a substrate loading and an enzyme/substrate loading of 0.65 wt% and 0.15 mM enzyme/g PA6 film, respectively.” We have additionally made the phrasing clearer in the “In-depth characterization” section to state that only one variable is being changed at once.

Line 195; “two squares of 0.5 x 0.5 cm PA6 film=13 mg” Please state the density of the material used (assuming 1.14 g/ml, the calculated mass would be 11.4 mg, not 13 mg)

We thank the reviewer for their comment. The mass was calculated by weighing six pairs of 0.5 x 0.5 cm squares and calculating the average of these measurements, not via a density calculation. This gave us an average mass of ~ 13 mg PA6 per reaction (13.19 mg \pm 0.13 mg). We believe small differences between ideal calculations versus what we observed could potentially be caused by a mixture of slight differences in thickness of the Goodfellow films, slight errors in the size of the squares, and small amounts of moisture still being absorbed into the film sheets despite drying. However, as shown by the small standard deviation of the mass measurements, any

differences causing deviation from an ideal calculated mass appear highly consistent across the set of squares, hence we believe weighing to be a suitable method for describing the mass of PA6 in our reactions.

To aid clarity we have updated the Methods section: “(13.19 mg \pm 0.13 mg PA6, calculated by weighing six pairs of squares and taking the average mass)”.

Figure 4; “NyCp2” should be “NylCp2”

We have updated the figure to read NylC_{p2}.

Line 215; list the enzymes/colors in the same order as in the figure

We have updated the figure caption to match the reviewer’s suggestion.

Line 220; “a grey circle” is used throughout the manuscript where “grey circles” would sound more appropriate

We have updated throughout the manuscript to “grey circles”.

Line 227; “70°C”; inconsistency regarding the space before the °C symbol, please check throughout the manuscript

We have updated all instances of this in the manuscript to have a space before the °C.

Line 239; “Despite the promise...” Here it is not clear whether the 0.67 wt% in “monomer release at 70 °C after 10 days only equates to 0.67 wt% PA6” refers to 6-AHA or 6-AHA equivalents, since NylCK-TS does not degrade the dimer (Figure S12C). The amount for 6-AHA equivalents would be of interest here

We thank the reviewer for their comment and agree the wrong word (monomer) was used here. The %wt depolymerization is worked out from the release of the soluble PA6 depolymerization products, converting the concentration of each oligomer in the sample to a mass of that oligomer in the sample using its associated molecular weight (the reviewer is correct that this can also be calculated from the 6-AHA equivalent value which gives the same mass).

We added the concentration of 6-AHA equivalents to the sentence and rephrased to aid clarity: “... the highest soluble PA6 oligomer release seen (339.7 μ M of 6-AHA equivalents) only equates to 0.67 wt% PA6 depolymerization extent.”

Line 260; “which produces only 49.8 μ M of 6-AHA” Here, again, it is unclear if “6-AHA” refers to the monomer or to “6-AHA equivalents” (looks like equivalents in Figure 4B)

Indeed, the word “equivalents” was mistakenly omitted.

We updated the sentence to “6-AHA equivalents”.

Line 271; “of over 10 days” > “over 10 days”

We have updated to “over 10 days”.

Line 277; “Supplementary Fig. 15” It is not clear from part A of the figure whether there is no data for day 10 for GatA+ or whether the dimer was undetectable. Same for GatA+ for part B of the figure, but for all days. For part A, please explain why dimer concentrations are increased on days 3 and 7, relative to day 1, while on day 1 it is much lower than in the no-enzyme control

We thank the reviewer for picking this up, there was indeed a plotting error in **Supplementary Fig. 15**, caused by a linked Excel file copying error, which was then also referred to in the main text. However, upon reflection we also believe that this was an unclear way to demonstrate the cyclic dimer hydrolysis activity of GatA. Hence, we ran additional experiments with a cyclic-dimer standard incubated with GatA, to demonstrate hydrolysis of this to monomer and dimer over time, to match the experiments conducted with the other small oligomers.

This is presented in **Supplementary Fig. 20**.

Figure 5; Part A: The legends on the left and right of the figure do not obviously represent the entire figure, it would be clearer if they were both moved to the right of the figure

We have moved the figure legend to the right of the panel as the reviewer suggested.

Line 318; “data in panel A and B” > “data in panels A and B”

We have updated the test to “data in panels A and B”.

Line 354; My most serious question is whether the accessible fraction is dependent on the surface area of the plastic used; it likely is. This leads to the question of why the authors did not investigate the effects of grinding or other means of increasing surface area (e.g., making nanoparticles) on the rate and extent of degradation of the polymer. While it is not necessary to add such experiments to the current paper, the issue must be discussed

We thank the reviewer for raising this interesting point. We have conducted powdered nylon reactions (using commercially available PA6 powder from Goodfellow), but this led to several challenges as the powder absorbs a significant amount of water during reaction, with the amount of water absorbed being inconsistent between no enzyme and enzyme-containing reactions. This makes the calculation of concentrations of products released very challenging, as the volume of the reactions is constantly reducing. There was also significant clumping and sticking of the powder to the reaction vessel walls, which led to highly heterogeneous reaction mixes. In addition, the commercially available powder from Goodfellow has a much higher crystallinity than the film (powder crystallinity = 47.6%) so we do not believe this would be a fair comparison to the more amorphous film.

We agree with the reviewer that the information on the Goodfellow powder may be useful for other researchers, hence we have included a line in the main manuscript to clarify our choice of a PA6 film for our experiments and a picture showing the dramatic water uptake of the nylon powder during reaction (**Supplementary Fig. 28**), alongside the associated characterization of the nylon powder in a new **Supplementary Table 5**.

Although increasing the surface area by using a powder was therefore complex, we tried to answer the reviewer’s point by instead decreasing the surface area of the film in the reaction by using a thicker film (Goodfellow PA6 film, 0.5 mm, 26% crystallinity). This decreases the surface area of 13 mg of film by around 50 %. Although the crystallinity is slightly higher than the main PA6 film used in the study, we believe this is a fairer comparison.

Reactions with NylC_K-TS confirm that the reduced surface area does indeed lead to a lower amount of product release, at a reduced rate with the data presented in **Supplementary Fig. 29** and discussed in the “Examination of the reaction profile of NylC_K-TS” section.

Line 411; doi.org/10.1038/s41467-018-07488-0 may be relevant to readers

We thank the reviewer for highlighting this paper, it is indeed highly relevant.

We added this reference to the manuscript.

Line 416; “0.18 mg enzyme/g PA6” is not in the same format as other amount enzyme/amount PA values in the manuscript, it would be easier to compare if it were

We have standardized the reporting of enzyme/substrate loadings throughout the manuscript.

Lines 427-431; Comparison to the wild-type would be very helpful here. That this enzyme (NylCK-TS), engineered in the current work, is the most active of the nylonases tested, increases the relevance of the paper to a journal like Nature Communications. However, the increase in performance (thermostability and 6-AHA production at various temperatures) must be clearly stated relative to the wild-type

As detailed above, in response to reviewers comments we have added a full dataset of comparisons of the two enzymes from 40-80 °C in **Supplementary Fig. 16**, and an additional main figure (**Fig. 5**) and associated text in the main manuscript.

Line 741; BL21(DE3)

We have updated the test to “BL21 (DE3)”.

Line 761; “concentration of 10X” is unclear

We thank the review for their comment. This is the normal nomenclature from the manufacture for stating the concentration of the dye stock in the assay (the manufacturer unfortunately does not provide any other measure of dye stock concentration).

However, we have added the stock concentration as provided by the manufacturer to aid clarity: "...SYPRO Orange dye (provided at a 5000X concentration from the manufacturer), added to a final concentration of 10X".

Line 766; Please check substrate name

This has been updated to *N*-(4-nitrophenyl)butanamide (N4NB) throughout the manuscript.

Lines 819-820; Please check enzyme amount, as commented on earlier in this report

We have standardized the reporting of enzyme/substrate loadings throughout the manuscript.

Line 832; "quenched in 5X MeOH" does this mean five volumes of methanol?

We thank the reviewer for their comment, and we agree this phrasing was ambiguous.

We have updated the methods to aid clarity: "Immediately following sampling, all time point samples were quenched in MeOH (1 volume sample: 4 volumes methanol) to denature any active enzyme, filtered using a 0.22 μ M PVDF filter plate to remove any solids, and stored at 4 °C prior to analysis."

Supplementary Fig. 1: Comparisons to the wild-type proteins would be valuable

We have added the three wild-type proteins to this figure.

Supplementary Fig. 2: Fix substrate name

This has been updated to *N*-(4-nitrophenyl)butanamide (N4NB) throughout the manuscript.

Supplementary Fig. 5: "grey circles"

We have updated throughout the manuscript to "grey circles".

Supplementary Fig. 12: "grey circles"

We have updated throughout the manuscript to "grey circles".

Supplementary Fig. 13: There are no grey circles in this figure

We have removed the reference to grey circles.

Supplementary Fig. 14: An AlphaFold model is not a "crystal structure"

We have updated the figure caption to "**AlphaFold predicted structure of NfPolyA**. The green ribbon represents the structure of NfPolyA as predicted using AlphaFold."

Supplementary Fig. 15: See comment to line 277

We have standardized the reporting of enzyme/substrate loadings throughout the manuscript.

Supplementary Fig. 16: Data points are represented by squares, triangles, and circles, not just circles

We have updated the figure caption to "circle, square, and triangle points represent the mean value of the triplicate measurements."

Supplementary Fig. 18: Investigation of the effect of increasing surface area would have been very interesting (grinding or making nanoparticles)

As detailed above, the commercially available PA6 powder does not have comparable characteristics to the PA6 film used in this study, hence we believe that would be an unfair comparison to our presented data. In addition, standardizing powder-based reactions due to water uptake appears complex and will require additional work which is outside the scope of this study.

As discussed in more detail above, we have added additional experiments with a thicker PA6 film to lower the surface area of the substrate to hopefully address this point in a more controllable way, showing that surface area is indeed important.

Reviewer #2 (Remarks to the Author):

In this article, the authors investigated the enzymatic hydrolysis of Nylon-6 (PA6). The reviewers posit that the authors have furnished readers with novel insights on the following aspects:

- (1) The authors achieved amino acid substitutions toward NylCK, distinct from the NylCp2 base GYAQ mutant extensively scrutinized by Negoro's group, with the aim of enhancing the heat resistance property.
- (2) The analysis encompassed enzymatic degradation of commercially available PA6 film.

Regrettably no novel superfamily was unveiled, though notwithstanding the authors' efforts to screen a new enzyme superfamily capable of hydrolyzing PA6 from the vast expanse of nature. While acknowledging the pioneering endeavors of Negoro's group in the realm of nylon hydrolytic enzymes, the authors ultimately discerned only the same NylB- and NylC-type enzymes. Furthermore, the degradation test for the PA6 films were on par with those previously reported by Negoro's group.

Hence, while the reviewer does not impugn the validity of the authors' work, the reviewer harbors reservations about whether the authors' findings constitute a sufficiently remarkable advancement to warrant publication in Nature Communications. The reviewer strongly recommends that, after the addition of biochemical characterization test for the newly discovered NylCK mutants, it may be judicious to resubmit the papers such as FEBS J or Sci. Rep. and so on.

We thank the reviewer for reviewing the manuscript. We disagree that the main findings of the paper can be summarized by points (1) and (2). We believe this to be the first attempt to compare known nylonases with consistent assay conditions and analysis, which are both extremely important to evaluate potential candidates for further engineering and study. We would like to refer Reviewer 2 to the comments made by Reviewer 1 and Reviewer 3, as to why studies such as this are important. Reviewers 1 and 3 additionally highlight the inconsistencies in the current literature with respect to both analysis methods and assay conditions, which makes comparing previous studies challenging.

Furthermore, we could find no evidence of NylBs being previously used on solid PA6 (which is reported in our manuscript), which may be useful for further investigation due to their production of a single monomer product. Additionally, previous studies have also not examined the product distributions seen while deconstructing PA6 with different enzyme types. Many of the enzymes we tested (e.g. GatA, ReU, LCC-ICCG to name a few) have not been previously reported for activity on solid PA6, thus we also disagree that no new enzymes were shown to be nylon-active. In addition, we comprehensively explore why NylC cannot fully deconstruct the PA6 film, which to our knowledge has not been reported previously, and is highly relevant for future studies as it appears that pre-processing the PA6 in some way prior to deconstruction is going to be essential.

We also disagree that the deconstruction analysis is the same as that has been previously used to examine NylCs. Using previously described methods, to our knowledge, there is no accurate quantification of multiple different products released following PA6 deconstruction. Here, we monitor 8 soluble products of PA6 deconstruction using a state-of-the-art LC-MS/MS method as highlighted by Reviewer 4. Previous methods have not looked at product release profiles in detail over time, and have instead commented upon released amine groups, which as we demonstrate here, will fail to capture the diversity of product release profiles.

In response to the reviewer's comment, we included an additional three enzymes in the study which are known urethanases, but which have not been previously reported to deconstruct PA6 (UMG-SP-1, UMG-SP-2, UMG-SP-3). Interestingly UMG-SP-1 acts like a NylB-type enzyme and the other two act more like the other amidases, further expanding our knowledge of known nylonases. These additional experiments and data can be found in **Figs. 1, 2 & 4, and Supplementary Figs. 2, 11,19, and 20**.

We are unsure as to what the reviewer is referring to when they suggest adding biochemical characterization of our engineered variant, NylC_K-TS. This is already extensively discussed in the paper. NylC_K-TS has been examined for its thermostability, it has been used to depolymerize solid PA6 in reactions from 40-70 °C, we explored how changing enzyme and substrate loading affected the reaction of NylC_K-TS, and additionally looked at pH effects on performance, and examined the limitations of the enzyme, identifying that it may undergo non-

specific binding events. We therefore believe that we have provided an excellent breadth of biochemical characterization for NylC_K-TS.

To include more data related to enzyme characterization, we added an additional figure to the paper (**Fig. 5**), and associated data to show how NylC_K-TS compares to the wild-type enzyme at varying temperatures (presented in **Supplementary Fig. 16**). These data are additionally discussed in the main manuscript. We have also included a predicted structure with the mutations mapped on to give more information about where the mutations are located structurally.

Additionally, to enhance the paper's caliber, the reviewer implores the authors to reevaluate the following:

(i) Negoro's group introduced a method for quantifying reaction rates employing thin-layered films of nylon 6 or 66 on 96-well plates. Astonishingly, this method was not cited in the paper, constituting a grave omission.

Reference: Nagai, Keisuke, et al. (2014), 'Enzymatic hydrolysis of nylons: quantification of the reaction rate of nylon hydrolase for thin-layered nylons', *Appl. Microbiol. Biotechnol.*, 98 (20), 8751-61.

We thank the reviewer for their comment, and we have cited this paper in the revised version. However, we respectfully disagree that this is a “grave omission”. There are a wide variety of methods used for nylon deconstruction assessment, so we did not want to get into extensive discussions about previous methods and their pros and cons in the Introduction, as this would be better discussed in a review paper.

However, in response to reviewer's comment about this, we have now included a brief overview of the types of methods that have been previously used in the Introduction: “For instance, while some studies use proxy measurements to evaluate nylon-active enzymes, such as looking for polymer surface changes via rising-height analysis,^{43,46} or using indicator compounds to identify the release of amine-containing molecules,⁴⁹⁻⁵¹ others use soluble substrate analogues to probe enzyme kinetics,⁵² or look for changes in thickness of self-cast nylon films following enzyme application.⁵³ However, the large diversity of experimental set-ups and substrates used across such studies, makes a direct comparison of previously described enzymes challenging.”

(ii) During the screening of nylon hydrolytic enzymes, the authors employed N-butyl-4-nitroaniline as the substrate, a compound markedly distinct from nylon 6 monomer unit structure. Consequently, the reviewer harbors doubts about the efficacy of this substrate for enzyme screening, as it may be more suitable for the screening of NylB-type enzymes (exo-type), but ill-suited for the screening of endo-type enzymes like NylC. The endo-type NylC enzymes do not recognize this substrate, and it remains uncertain whether effective screening has been achieved. Demonstrating the validity of this screening system is imperative. From this perspective, the reviewer strongly suggests that adopting the aforementioned thin-layered film screening system could potentially unearth previously concealed enzymes.

We thank the reviewer for their comment. However, we believe there may have been a number of misunderstandings of the results presented. The reviewer states that NylC does not recognize the small molecule N4NB that we used for initial screening. The data in **Fig. 2** of the main text and **Supplementary Fig. 2**, clearly show that NylC-type enzymes can hydrolyze this substrate to release *p*-nitroaniline. **Supplementary Fig. 2** additionally shows the full hydrolysis reaction for the NylC-type enzymes over 24 hrs; the no enzyme control reaction shown in **Supplementary Fig. 2A** shows no hydrolysis of N4NB in just buffer, hence all the hydrolysis activity seen can be attributed to the NylC enzymes. This is also clearly described in the main text.

Secondly, we have already validated this system in the paper by testing all the enzymes on solid PA6 film.

We have now additionally provided the data for the protease enzymes in **Supplementary Fig. 12**.

Hence, it is unclear to us how enzymes could have concealed activity on PA6 film. We have characterized all the enzymes, looking for multiple released products, at different reaction temperatures, over long periods of time on a solid PA6 film substrate. We describe in the Discussion that while the N4NB assay is not a good indicator of reaction extent on solid PA6, it can be useful for discarding enzymes that may be inactive on PA6. We further highlight that coupling small molecule assays with more extensive solid PA6 assays is highly important.

The reviewer suggests that using a previously described method using the imaging of self-cast nylon thin films in 96-well plates would have provided a better screening system. We agree that the thin film-based method may be

a perfectly reasonable and useful approach for some applications, but we would like to provide the reviewer with more information as to why we believe this approach would not have been suitable to achieve our goals for this manuscript:

(1) Self-cast thin films may not accurately represent PA6 present in plastic waste as they will have unusual morphologies and characteristics (they are only a few 100 polymer chains thick). In the manuscript, we were aiming to identify enzymes in the context of future industrial application or engineering, hence, focusing on ultra-thin films may limit broader substrate compatibility.

(2) Scaling up ultra-thin film production from PA6 waste is not feasible for real-world applications, hence, optimizing for this type of substrate and trying to work out limitations of the enzymes on this substrate would not have been useful for future applications. We used a cut up PA6 film, which is scalable, practical, and representative of PA6 films, so this should give greater insight into problems that may be encountered when trying to scale the depolymerization process, as we discuss in the manuscript.

(3) We believe standardization and accessibility is key to promoting good comparisons between polymer deconstructing enzymes (see discussion in Ellis *et al.*, *Nature Catalysis* 2021). Hence, we chose a commercially available substrate to ensure reproducibility by other researchers and allowing us to conduct comprehensive substrate characterization. Self-casting films has the potential to be highly variable between different laboratories.

(4) In addition to substrate accessibility concerns, we also wished to create an analysis method that was widely accessible. The camera and imaging set up for the thin-film analysis is quite a specialized set up that we do not have access to. Hence, we used a common piece of analytical equipment (LC-MS/MS); as highlighted by Reviewer 1, such equipment is broadly available, and hence accessible to a large portion of the research community.

(5) Using pre-cut PA6 film allows flexible reaction scaling and variation in PA6 mass, which would be challenging using cast thin films in plates, due to surface area limitations.

(6) The LC-MS/MS method developed here provides far more detailed information on product release than the TNBS assay mentioned in the cited thin-film paper. Furthermore, the LC-MS/MS method adopted here is not affected by reaction times, buffer conditions, or product instability as is seen in the TNBS assay.

Additional data demonstrate our method can be used for lysates, as presented in Supplementary Fig. 7.

Therefore, although the thin-film method may be useful for comparing similar enzymes in a plate-based format, it would not have given the level of detail and consistency we were aiming to achieve here when comparing a highly variable enzyme panel. Additionally, results using the thin-film method would still have needed to be further re-evaluated using realistic substrates and more detailed characterization methods. Stated differently, we do not see the value of adding the thin-film assay as an additional step when we have detailed information using a solid PA6 film already. Thus, we posit that our approach has numerous benefits, and that the workflow and analysis used here together represent a separate and important addition for assessing nylonases, which has allowed us to determine more detailed information on the action of nylonases than has been described previously to our knowledge.

(iii) The authors are urged to validate the amino acid numbering of NylC mutants. For instance, S112G should be corrected to S111G, and E262Q should similarly be adjusted to E263Q.

We thank the reviewer for raising this point, we had numbered the amino acids to reflect the amino acid sequences provided in **Supplementary Table 2**, however we have updated the numbering to instead reflect the crystal structure numbering from PDB 3AXG. **The mutations are now listed as S111G, A137L and E263Q.**

Reviewer #3 (Remarks to the Author):

This manuscript by Bell and coworkers reports a comprehensive and systematic investigation of the enzymatic degradation of nylon. The work is well motivated and presents a high level of novelty. Thus, based on the chemical structure of Nylon, bioprocessing of Nylon waste indeed represents a promising case – particularly in the light of recent progress within enzyme-based monomer recovery from polyester waste. Yet, the literature on nylon hydrolases is sparse and quite incompatible with respect to methods and strategies.

To the best of my knowledge this work is the first systematic study where a larger group of potential nylon hydrolases have been identified, organized in different groups, produced, engineered, and characterized in a systematic way for direct comparison. This represents a huge work and an important step forward in our understanding of enzymatic nylon degradation.

The manuscript is clearly written with a comprehensive empirical material presented in compressed but well-composed figures as well as a substantial supplementary material.

I recommend the publication of this work.

We thank the reviewer for their positive feedback and their concise summary of the need for studies with standardized analysis to compare nylonases. We fully agree that these types of studies are important to aid future nylonase research efforts. We agree that variabilities in assay setup and analytical methodologies make comparing the current literature challenging. We too believe this to be the first study comparing many different potential nylonases in a standardized and systematic format.

A few comments for consideration.

Line 188: I guess the authors mean differential scanning calorimetry.

This was a mistake – we have updated this to differential scanning calorimetry.

Line 235-240. Are the oligomers discussed here stable in enzyme-free solutions or could the distribution of the different oligomers be affected by non-catalyzed hydrolysis?

We thank the reviewer for raising this interesting point and agree this information would be useful to others and would serve as nice control experiments. We therefore conducted additional experiments incubating known concentrations of the linear oligomers at the different reaction temperatures used in the study. As a note, our data already show that the cyclic products remain constant throughout the reactions (**Supplementary Fig. 5B**).

We are pleased to report that the linear oligomers seem fairly stable in solution, and we have presented these data in **Supplementary Table 4**. We have added an extra line in the “Assessment of potential nylonases” section to highlight these findings: “The linear oligomers (from monomer to trimer) appear robust to incubation for extended time periods at 40-80 °C, with an average recovery (as measured by LC-MS/MS) across all time points and temperatures of $96.4\% \pm 9.2$, $99.9\% \pm 6.3$ and $104.3\% \pm 7.2$ of individually incubated $10 \mu\text{g mL}^{-1}$ standards of 6-AHA monomer, 6-AHA dimer and 6-AHA trimer, respectively (**Supplementary Table 4**). These results indicate that changes in oligomer levels seen in reactions can be attributed mainly to enzyme action.”

Line 259: I did not understand what was meant by “concerted depolymerization”.

We clarified this sentence to read: “The reaction profiles and product distributions of all of the biocatalysts tested from this group suggest a similar mode of action, with the mixture of soluble oligomers released indicating non-specific cleavage of surface residues, rather than coordinated, progressive depolymerization activity”.

Line 333: I found it remarkable that BSA (at a 10:1 or so ratio to the enzyme) increases the specific activity dramatically (some 100-fold). The interpretation (lines 334-5) is: “that while NylCK-TS may be affected by non-specific binding events, this is not a main factor”. That seems like an understatement to me. If the effect of BSA is related to nonspecific adsorption to the reaction vial or non-productive binding to the PA surface, one could argue that the remaining enzyme population was two order of magnitude more active than what is captured by average numbers. I might be that I have misunderstood this, but if not, this would be an observation to bring on in future attempts to find or engineer better nylon hydrolases. I suggest the authors elaborate on this and its relationship e.g. to the dose response curves in Fig. 5B.

We thank the reviewer for their comments on this section and agree the phrasing of this section could be improved to aid clarity. We were trying to convey that potential non-specific binding events were not a main factor influencing the extent of depolymerization i.e. it appears if we add BSA, we do not observe any further depolymerization compared to a normal reaction. However, the reviewer is correct in that it does dramatically change the enzyme specific activity. We agree that the findings of the BSA experiments are highly interesting and agree that these findings could be more strongly highlighted to aid future engineering efforts. We have therefore elaborated on these findings and possible explanations in the main text and conclusion. We also agree another explanation could be that the enzymes can bind to reaction containers, and so have added this factor in.

We have moved the BSA assays to their own paragraph in the “Examination of the reaction profile of NylC_K-TS” section to improve clarity and have added additional explanation as to what might be occurring in the BSA containing reactions, highlighting that this will be important to consider in future enzyme engineering efforts. We have also added an additional sentence to the discussion highlighting the BSA findings: “Our experiments additionally suggest that NylC_K-TS activity may be substantially affected by non-productive binding events, indicating that this may be an important area for further investigation when considering future enzyme engineering attempts.”

Reviewer #4 (Remarks to the Author):

I was tasked with examining the applied mass spectrometric protocols employed to quantify the soluble fractions resulting from the enzymatic PA6 degradation. The authors have developed a careful protocol based on direct calibration of available standards. The used methodology is in line with best practice in the field and I could not identify any critical flaws or weaknesses in their approach.

We thank the reviewer for their concise summary of our analysis approach. We agree that it aligns with best practices in the field and hope our protocol.io methods walkthroughs will also be useful for helping other researchers conduct the developed technique.

Reviewers' Comments:

Reviewer #1:

Remarks to the Author:

I appreciate the care the authors have taken in responding to my numerous comments and I am satisfied with the changes they have made to the manuscript in response to my review. I would say the authors also appropriately responded to the comments made by the other reviewers and I agree with the changes made to the manuscript. There are only a few topics I would like to briefly touch on.

I appreciate the detail in which the authors responded to one of my comments about "enzyme concentration as a function of the mass of PA6". It seems I miscommunicated since all I was trying to say was that expressing the amount of enzyme used as 'concentration of enzyme' per 'mass of PA' could potentially be misleading for a surface area-limited process, where increasing the mass of plastic could be achieved without significantly altering surface area (e.g., by using a thicker film, as in fact the authors did in response to my question about using nanoparticles). I did not intend for the authors to change their units or to also express the plastic loading as wt%, although the latter is not necessarily unwelcome. Please note that in the process of changing the units, it seems like a conversion error was made in line 270 (it seems like the $\mu\text{M/g}$ should be $\mu\text{M/mg}$ or mM/g).

I am satisfied with the response the authors gave to my comment about not testing nanoparticles and the information they added to the manuscript will help other researchers decide to avoid nanoparticle assays too or to come up with solutions to the problems. I do however not think I understood the "picture showing the dramatic water uptake of the nylon powder during reaction" (Figure S28). The way it is currently phrased makes it sound like the total volume of the reactions decrease upon addition of 13 mg of PA nanoparticles to 2 ml of liquid, but the total volume should not decrease as water is absorbed by the plastic. Please clarify this addition to the manuscript to avoid confusion.

I appreciate that the authors have now provided the data for the protease enzymes in Figure S12. We have long been curious about the poor performance of proteases on polyamides but obviously polypeptides are structurally significantly different from nylon. The authors were correct in thinking that N4NB (p-nitrophenyl butyramide) would be a good substrate to test for initial activity, and in fact one can assay 'endo' cleaving hydrolases using small chromogenic substrates. An example from the literature is that p-nitrophenyl butyrate, an ester analogous to N4NB, is routinely used to assay cutinase activity, even though these enzymes are polymer hydrolases by nature (cutinase activity on pNPB is rather high, making this a sensitive and convenient assay system). Based on the data presented in this paper and previous work, I agree with the statement that "while the N4NB assay is not a good indicator of reaction extent on solid PA6, it can be useful for discarding enzymes that may be inactive on PA6" despite the subtle contradiction it expresses. The reason it is important to present both assays in this paper is because, even though the LC-MS/MS assay described in the manuscript is high-throughput, even higher throughput is needed if large mutant or natural diversity libraries need to be screened, and for this purpose a colorimetric plate reader-based assay is valuable. In that sense, using N4NB is much simpler than using self-cast nylon films and colorimetric PA6 assays. New enzymes found can then be directly compared to most previously described nylonases by using the LC-MS/MS assay presented in this manuscript.

The title of Supplementary Fig. 21, "Reactions of GatA with 6-AHA cyclic-dimer", seems to be incorrect (data is about trimer hydrolysis). In a few places, 'cystine' should be 'cysteine'.

Reviewer #2:

Remarks to the Author:

Appreciation is extended for authors' considerate and responsive reply addressing reviewer's queries.

The reviewer does not dispute the merit of the authors' endeavors. The reviewer recognizes the authors' commendable efforts in establishing a novel assay system for nylon hydrolysis and conducting a comprehensive exploration of the hydrolytic activity inherent in nylon amide bonds against various enzymes.

However from first submission, the reviewer is raising concerns regarding the authors' failure to adequately introduce the ongoing research conducted by Negoro et al. on the hydrolysis of polymeric nylon. Notably, it was elucidated in ref. 49 that a NylC mutant weakly but clearly exhibited the hydrolytic activity on bulk nylon particles. However, Reviewer feels that the authors still did not accurately introduce this point in the main body of the manuscript.

The reviewer believes that this point should be remedied prior the publication of this article. Truly so sorry, but the reviewer requests earnest reconsideration on this matter.

**Expressions of gratitude for authors' attention to this concern.
Thank you very much.**

Reviewer #3:

Remarks to the Author:

The authors have addressed the criticism in the original report in a satisfactory manner. I recommend the publication of this work in its current form.

Reviewer comments are provided in black font. Our responses are shown in blue font. Changes to the manuscript are shown in red font.

Reviewer #1 (Remarks to the Author):

I appreciate the care the authors have taken in responding to my numerous comments and I am satisfied with the changes they have made to the manuscript in response to my review. I would say the authors also appropriately responded to the comments made by the other reviewers and I agree with the changes made to the manuscript. There are only a few topics I would like to briefly touch on.

We thank the reviewer for taking the time to review the updated manuscript and for their very close attention to detail on both the original and revised manuscript.

I appreciate the detail in which the authors responded to one of my comments about “enzyme concentration as a function of the mass of PA6”. It seems I miscommunicated since all I was trying to say was that expressing the amount of enzyme used as ‘concentration of enzyme’ per ‘mass of PA’ could potentially be misleading for a surface area-limited process, where increasing the mass of plastic could be achieved without significantly altering surface area (e.g., by using a thicker film, as in fact the authors did in response to my question about using nanoparticles). I did not intend for the authors to change their units or to also express the plastic loading as wt%, although the latter is not necessarily unwelcome. Please note that in the process of changing the units, it seems like a conversion error was made in line 270 (it seems like the $\mu\text{M}/\text{g}$ should be $\mu\text{M}/\text{mg}$ or mM/g).

We thank the reviewer for clarifying their comment. However, we believe the standardization of units across the manuscript was still an improvement, even if that was not the reviewer’s intention. While we agree that the PA6 deconstruction is a surface area-limited process, we have decided to use the “standard” reporting units for reactions with a solid substrate including an enzyme substrate loading (enzyme per mass of substrate) and substrate loading (both as mass and as a wt%) as this appears to be the standard the community is converging on for enzymatic depolymerization reactions (as in <https://pubs.acs.org/doi/epdf/10.1021/acscatal.3c02922>, ref 54 in the manuscript), and we would like to keep this description of reactions as good practice. Moreover, this is the unit that is directly useful for estimating process costs for enzymes, as is commonly reported in the cellulase community (see, e.g., <https://pubs.acs.org/doi/full/10.1021/cr500351c>).

Additionally, we think the described metrics are a good way to describe a standard reaction set up, which can also be easily replicated. While stating enzyme loading per unit surface area may be possible for this study, it may make for difficult comparisons if others wish to compare results with alternative substrate forms such as shredded films or powders where a surface area calculation may be challenging. We believe the experiments in the manuscript and associated text clearly demonstrate the surface area limited nature of the process so we anticipate that this will not be misleading to readers.

In response to the reviewer’s comments, we have added an additional line to the “Development of high-throughput solid PA6 depolymerization” to explain this choice and avoid any ambiguity:

“Although polymer deconstructions may be surface area-limited processes, the enzyme/substrate loading is referred to throughout as amount of enzyme per mass of PA6 as is good practice for enzymatic depolymerization reactions.^{54,56,85”}

We thank have corrected the units in line 270 to “mM enzyme/g PA6”.

I am satisfied with the response the authors gave to my comment about not testing nanoparticles and the information they added to the manuscript will help other researchers decide to avoid nanoparticle assays too or to come up with solutions to the problems. I do however not think I understood the “picture showing the dramatic water uptake of the nylon powder during reaction” (Figure S28). The way it is currently phrased makes it sound like the total volume of the reactions decrease upon addition of 13 mg of PA

nanoparticles to 2 ml of liquid, but the total volume should not decrease as water is absorbed by the plastic. Please clarify this addition to the manuscript to avoid confusion.

We agree with the reviewer that the uptake of water by the nylon powder is an interesting phenomenon and agree that the phrasing may have been confusing. We believe this to be a complex process involving the nylon powder absorbing water in addition to enzyme action causing a type of “excess volume” effect, and indeed, the uptake of water by nylon-6 is a known issue that affects its performance (<https://doi.org/10.1016/j.xcrp.2022.100840>). This could be due to the particles/reaction products affecting the organization of water molecules, or the absorption of the water into the nylon structure leading to a more compact configuration of the water molecules leading to volume contraction (<https://doi.org/10.1080/15583724.2020.1855196>). It has been previously noted that enzyme hydrolytic reactions can cause reaction volume reductions (*Biochem J* (1929) 23 (5): 975–981), so this may also be a factor. As a complete understanding of this process will require further experimentation, we have updated the text to remove ambiguity and instead highlight that this is a feature of enzymatic powder reactions rather than trying to infer the method by which this happens. The text now reads:

“Typically, this is done by using a powdered substrate; however, we found that enzymatic assays with PA6 powder (sourced from Goodfellow, particle size: 5-50 μm , full material characterization detailed in **Supplementary Table 5**) were difficult to standardize due to reaction volume changes over the course of incubation (**Supplementary Fig. 28**).”

I appreciate that the authors have now provided the data for the protease enzymes in Figure S12. We have long been curious about the poor performance of proteases on polyamides but obviously polypeptides are structurally significantly different from nylon. The authors were correct in thinking that N4NB (p-nitrophenyl butyramide) would be a good substrate to test for initial activity, and in fact one can assay ‘endo’ cleaving hydrolases using small chromogenic substrates. An example from the literature is that p-nitrophenyl butyrate, an ester analogous to N4NB, is routinely used to assay cutinase activity, even though these enzymes are polymer hydrolases by nature (cutinase activity on pNPB is rather high, making this a sensitive and convenient assay system). Based on the data presented in this paper and previous work, I agree with the statement that “while the N4NB assay is not a good indicator of reaction extent on solid PA6, it can be useful for discarding enzymes that may be inactive on PA6” despite the subtle contradiction it expresses. The reason it is important to present both assays in this paper is because, even though the LC-MS/MS assay described in the manuscript is high-throughput, even higher throughput is needed if large mutant or natural diversity libraries need to be screened, and for this purpose a colorimetric plate reader-based assay is valuable. In that sense, using N4NB is much simpler than using self-cast nylon films and colorimetric PA6 assays. New enzymes found can then be directly compared to most previously described nylonases by using the LC-MS/MS assay presented in this manuscript.

We thank the reviewer for their summary of the importance of including both methods.

The title of Supplementary Fig. 21, “Reactions of GatA with 6-AHA cyclic-dimer”, seems to be incorrect (data is about trimer hydrolysis). In a few places, ‘cystine’ should be ‘cysteine’.

We thank the reviewer for identifying this mistake. The graph is indeed about the cyclic-dimer experiments, but one of the graph color key labels was incorrectly named as “trimer” instead of “cyclic-dimer”, which has caused the confusion.

We have updated the color key label to “cyclic-dimer” of Supplementary Fig.21 accordingly.

Reviewer #2 (Remarks to the Author):

Appreciation is extended for authors' considerate and responsive reply addressing reviewer's queries.

The reviewer does not dispute the merit of the authors' endeavors. The reviewer recognizes the authors' commendable efforts in establishing a novel assay system for nylon hydrolysis and conducting a comprehensive exploration of the hydrolytic activity inherent in nylon amide bonds against various enzymes.

However from first submission, the reviewer is raising concerns regarding the authors' failure to adequately introduce the ongoing research conducted by Negoro *et al.* on the hydrolysis of polymeric nylon. Notably, it was elucidated in ref. 49 that a NylC mutant weakly but clearly exhibited the hydrolytic activity on bulk nylon particles. However, Reviewer feels that the authors still did not accurately introduce this point in the main body of the manuscript.

The reviewer believes that this point should be remedied prior the publication of this article. Truly so sorry, but the reviewer requests earnest reconsideration on this matter.

We thank the reviewer for taking the time to review the updated manuscript and welcome their positive comments. We appreciate the previous work conducted by the Negoro group, as is reflected by the citation of numerous Negoro papers throughout the manuscript. We would like to draw the reviewer's attention to several examples where the previous work of Negoro *et al.* is highlighted:

- the discovery and use of NylCs to deconstruct PA6 by Negoro *et al.* is mentioned in lines 44-49 of the Introduction with the associated references for the Negoro *et al.* papers.
- the thin film analysis method by Negoro *et al.* is also mentioned and referenced in lines 57-60.
- the mutation of the NylCs for thermostability is additionally included in the manuscript, as clearly referenced and described in lines 131-140, where Negoro *et al.* is mentioned by name.
- previous work by the Negoro group is also referenced in the discussion (lines 502-504).

We do not state at any point in the manuscript that we are the first to try NylCs on solid PA6, as described in the "Identification, expression, and amide bond hydrolysis assays of potential nylonases" section, wherein we note that previously described activity on polymeric PA6 and its oligomers was one of the criteria we used to select enzymes in the first place.

We do not feel that there is scope or need within this original research paper to do an in-depth review of all previous work on solid PA6 deconstruction by enzymes, and instead we have provided comprehensive tables (**Supplementary Tables 1-2**), which have the source of each enzyme used in the paper, which polymers it has been previously shown to have activity on, and the associated references, so readers can easily find the original research.

Regardless, to address to the reviewer's comments, we have added an additional line to the "Identification, expression, and amide bond hydrolysis assays of potential nylonases" section to more strongly highlight the previously described capabilities of NylCs (additionally already detailed in **Supplementary Table 1**):

"As demonstrated by the Negoro *et al.*, NylC_{p2} and its homologues (Ntn hydrolases) have previously been shown to exhibit activity on PA6 cyclic oligomers, linear oligomers and solid PA6 (powder and thin film),^{24,37,49,53,63} whilst NylB and its homologues have been shown to exhibit PA6 dimer hydrolysis activity.^{38,39}"

We have also added that the described NylC mutant constructed by Negoro *et al.* had activity on solid PA6 powder, and included additional details of this mutant:

"Inspired by the successful homology-based thermostability enhancement of NylC_{p2} by Negoro *et al.* (NylC_{p2}D122G/H130Y/D36A/E263Q, T_m increase of 36 °C, demonstrated hydrolytic activity on solid PA6 powder),⁴⁹ we also rationally mutated NylCs from *Agromyces* (NylC_A) and *Kocuria* (NylC_K) to increase their melting temperatures (T_ms).^{37,49,66}"

In ref. 49, unfortunately there is no comparison of the solid PA6 hydrolysis activity of the mutant to the WT enzyme, else we would have also included this comparison in this sentence.

We believe with the already included citations and descriptions of the work therein, and the updated sentences in mind, the previous work on NylCs is adequately highlighted. Our manuscript is examining the nylonase activity of a diverse range of enzymes previously described by many groups, hence we feel that including any more detail on just one class of enzymes would take away from the main aims of the paper or would require additional reviews of all the enzyme types tested, which is outside the scope of the manuscript.

Expressions of gratitude for authors' attention to this concern.

Thank you very much.

Reviewer #3 (Remarks to the Author):

The authors have addressed the criticism in the original report in a satisfactory manner. I recommend the publication of this work in its current form.

We thank the reviewer for reviewing the updated manuscript.